# EvoMAS: Evolutionary Generation of Multi-Agent Systems

**Yuntong Hu** [1][†]  **Yuting Zhang** [2][*]  **Matthew Trager** [2]  **Yi Zhang** [2]  **Shuo Yang** [2]  **Wei Xia** [2]  **Stefano Soatto** [2]

## Abstract

Large language model (LLM)–based multi-agent systems (MAS) show strong promise for complex reasoning, planning, and tool-augmented tasks, but designing effective MAS architectures remains labor-intensive, brittle, and hard to generalize. Existing automatic MAS generation methods either rely on code generation, which often leads to executability and robustness failures, or impose rigid architectural templates that limit expressiveness and adaptability. We propose Evolutionary Generation of Multi-Agent Systems (EvoMAS), which formulates MAS generation as structured configuration generation. EvoMAS performs evolutionary generation in configuration space. Specifically, EvoMAS selects initial configurations from a pool, applies feedback-conditioned mutation and crossover guided by execution traces, and iteratively refines both the candidate pool and an experience memory. We evaluate EvoMAS on diverse benchmarks, including BBEH, SWE-Bench, and WorkBench, covering reasoning, software engineering, and tool-use tasks. EvoMAS consistently improves task performance over both human-designed MAS and prior automatic MAS generation methods, while producing generated systems with higher executability and runtime robustness. EvoMAS outperforms the agent evolution method EvoAgent by +10.5 points on BBEH reasoning and +7.1 points on WorkBench. With Claude-4.5-Sonnet, EvoMAS also reaches 79.1% on SWE-Bench-Verified, matching the top of the leaderboard. Code is available at https://github.com/amazon-science/EvoMAS.

[†]Work done during an internship at AWS AI. [*]Corresponding author. [1]Department of Computer Science, Emory University, Atlanta, GA, USA [2]AWS, New York, NY, USA. Correspondence to: Yuting Zhang <yutingzh@amazon.com>.

*Proceedings of the 43rd International Conference on Machine Learning*, Seoul, South Korea. PMLR 306, 2026. Copyright 2026 by the author(s).

## 1. Introduction

Agents based on large language models (LLMs) have demonstrated strong capabilities in reasoning, planning, and coding (Stein et al., 2025; Yao et al., 2022; Wang et al., 2024). However, many real-world problems require coordinating multiple subtasks with different tools, iterative feedback from dynamic environments, and context organization across asynchronous settings (Chen et al., 2024b; Hong et al., 2024; Pan et al., 2025). While single agents can solve isolated subtasks, they struggle with long-horizon coordination, systematic verification, and orchestrating heterogeneous tools across interdependent subtasks. Multi-agent systems (MAS) address these challenges by decomposing complex tasks into interacting agents with specialized roles, tool access, and coordination patterns (He et al., 2025; Kim et al., 2025; Shu et al., 2024). In this work, we focus on collaborative MAS organized in hierarchical and peer-to-peer structures.

Despite their effectiveness, designing multi-agent systems remains labor-intensive and error-prone, requiring expert knowledge to specify agent roles, interaction structures, and execution logic (Zhou et al., 2025; Chen et al., 2024b; Hong et al., 2024; Qian et al., 2024; Zhuge et al., 2024; Liu et al., 2024). Manual design does not scale across tasks and often fails to adapt to diverse or evolving requirements (Cemri et al., 2025).

To address this limitation, a few works explore automatic generation of MAS using LLMs, aiming to construct task-specific agent systems with minimal human intervention. Existing approaches differ fundamentally in how agent systems are represented and manipulated, ranging from implicit natural-language descriptions and procedural code to explicit architectural templates and schemas. These representation choices strongly affect the scalability, robustness, and adaptability of automatically generated MASs. **AutoAgents** (Chen et al., 2023) generates agent roles and execution plans through a predefined LLM pipeline, but relies on hand-designed frameworks and represents MAS only implicitly via natural language. **ADAS** (Hu et al., 2024) formulates MAS design as open-ended code search, but lacks explicit configuration-level abstraction. **MAS-GPT** (Ye et al., 2025) directly generates executable multi-agent programs in a single forward pass, but relies on fine-tuning with curated code,

limiting generality. **EvoAgent** (Yuan et al., 2025) adopts an evolutionary approach by mutating agent prompts and roles, but evolves individual agents rather than treating the multi-agent system as the optimization object.

Existing approaches for generating executable MAS expose a design tension between expressiveness and reliability. Methods such as MAS-GPT (Ye et al., 2025) and ADAS (Hu et al., 2024) emphasize broad exploration through code generation but often suffer from executability failures and brittle behaviors. In contrast, AutoAgents (Chen et al., 2023), EvoAgent (Yuan et al., 2025), and MetaAgent (Zhang et al., 2025b) improve execution reliability by constraining agent structures, at the cost of reduced generality.

To address the above challenges, we introduce a configuration-based paradigm that optimizes structured MAS configurations. Each configuration declaratively defines agent prompts, tool access, backbone models, and inter-agent topology, executed by a lightweight runtime interpreter that decouples system structure from execution logic. Configuration generation proves more robust than direct code generation while enabling broad exploration of the MAS design space. This approach aligns well with today's LLM capabilities, enabling efficient inference-time adaptation.

Building on this paradigm, we propose **Evo**lutionary Generation of **M**ulti-**A**gent **S**ystems (**EvoMAS**), a method that evolves collaborative MAS configurations via selection, mutation, and crossover. The MAS generator uses execution feedback to iteratively refine candidates and maintain a pool seeded from human-designed systems across multiple domains. EvoMAS works with different base agents, such as Smolagent and SWE-Agent, making it a generalizable and agent-agnostic framework.

Evolutionary MAS generation is a promising way for test-time compute scaling (Snell et al., 2025). Rather than allocating compute within a single agent, evolution adaptively distributes compute across the multi-agent system through search over configurations. Evolution also provides a state-based learning paradigm where improvements accumulate through configuration updates and execution experience, distinct from gradient-based fine-tuning.

Our core contributions are:

- A paradigm that reformulates MAS construction as structured text generation for flexible configurations, enabling robust and expressive MAS generation.

- EvoMAS, an LLM-driven evolutionary framework that maintains a pool of candidate configurations seeded from human-designed MASs and iteratively refines them via execution-guided mutation, crossover, and memory-based reuse.

- Benchmarking across reasoning, coding, and tool-calling tasks demonstrating stronger task performance across domains with higher execution success rates.

**Conflict of Interest Disclosure.** All authors were employed by Amazon Web Services (AWS) at the time of this research. The experiments use LLM APIs accessed through AWS Bedrock and locally hosted open-source models. AWS did not impose constraints on the experimental design, analysis, or conclusions of this work. The evaluation benchmarks are publicly available and developed independently of AWS.

## 2. Problem Formulation

**Definition 2.1** (Multi-Agent System Configuration). A *multi-agent system (MAS) configuration* is defined as a tuple $C = (G, \{A_i\}_{i=1}^k, V_{\text{in}}, V_{\text{out}})$. The component $G = (V, E)$ is an acyclic communication graph with agent set $V = \{v_1, \ldots, v_k\}$ and directed edges $E \subseteq V \times V$; an edge $(v_i, v_j) \in E$ specifies that the outputs of $v_i$ are (part of) the inputs of $v_j$. For each agent $v_i$, $A_i = (b_i, p_i, \Gamma_i)$ denotes its *agent configuration*, consisting of a backbone model $b_i$, a system prompt template $p_i$, and a tool set $\Gamma_i$. Finally, $V_{\text{in}}, V_{\text{out}} \subseteq V$ are distinguished subsets specifying the input and output agents, respectively.

Given an MAS configuration $C$ and task description $q \in \mathcal{Q}$, *executing* $C$ on $q$, denoted $\text{Exec}(C, q)$, proceeds by: (1) providing $q$ as input to all agents in $V_{\text{in}}$, (2) allowing agents to communicate according to $E$ until a termination condition is met where the head agent passes the accumulated context to the tail agent along the corresponding edge, and (3) collecting outputs from agents in $V_{\text{out}}$.

Let $R : \mathcal{Q} \times \mathcal{C} \to \mathbb{R}$ be a reward function that evaluates how well $\text{Exec}(C, q)$ solves task $q$. Given a task $q \sim \mathcal{Q}$, we aim to find a configuration $C_q^*$ that maximizes the reward $R(q, C)$. The configuration selection should benefit from experience: observing the performance of configurations on prior tasks should inform choices for new tasks.

This formulation introduces several challenges:

- How to efficiently explore the large and combinatorial configuration space?

- How to leverage execution feedback to guide structured configuration updates?

- How to balance exploration of new configurations with reuse of successful ones?

## 3. Methodology

To address these challenges, we propose **Evolutionary Generation of Multi-Agent Systems (EvoMAS)**, which au-

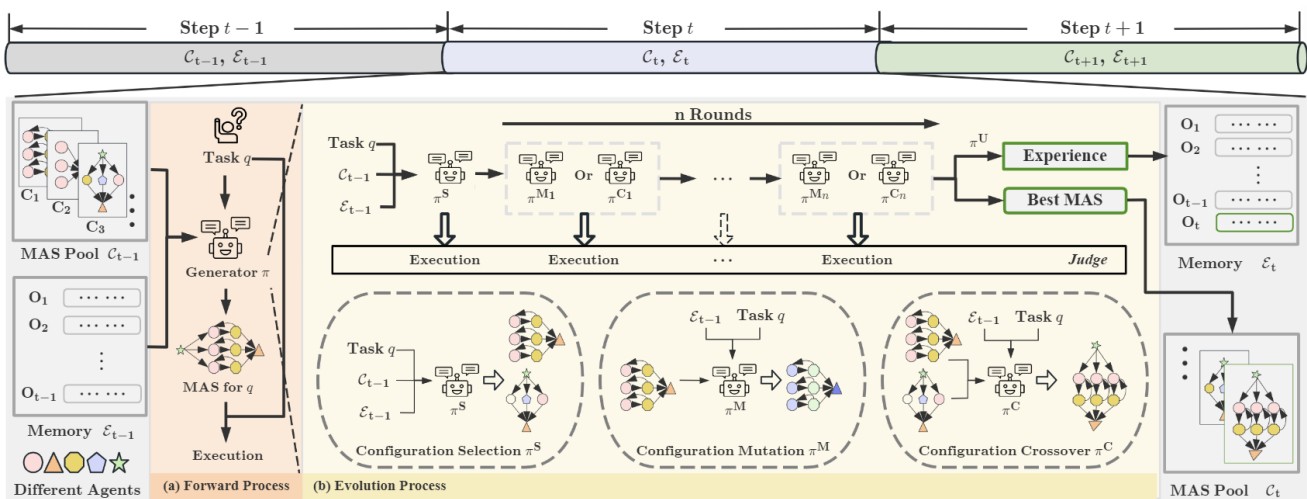

*Figure 1.* Overview of EvoMAS. Given a task, the MAS generator produces structured configurations specifying agent roles, model assignments, prompts, and communication topology. The MAS executor instantiates agents accordingly and executes the task. A verifier evaluates outputs to compute reward signals, which guide evolutionary optimization through mutation and selection over multiple generations.

tomatically generates MAS configurations for each task. EvoMAS uses an LLM to guide evolutionary search over a configuration candidate pool, using execution feedback as learning signals. The approach does not assume access to ground truth during configuration selection, and instead uses an LLM-as-judge metric as a proxy reward signal.

For each incoming task, EvoMAS samples initial configurations from the pool, evolves them through LLM-guided mutation and crossover operators, and selects the best configuration for final execution. The system improves over time by (1) retaining successful configurations in the pool for reuse, and (2) storing summaries of evolution traces in memory to guide future search.

### 3.1. Overview

EvoMAS operates on structured MAS configurations that specify agent roles, backbone models, system prompts, tool access, and communication topology (see Appendix B for the full schema and details).

We borrow the conceptual framework from genetic algorithms (Goldberg, 1989; Li et al., 2023b), but unlike traditional approaches that use random perturbations, EvoMAS leverages LLM-driven operators conditioned on task context and execution feedback. As shown in Figure 1, we define four operators:

- Select: retrieves and instantiates candidate configurations from the pool based on task relevance;

- Mutate: locally refines a single configuration by editing agent specifications, prompts, or topology, conditioned on

execution feedback;

- Crossover: recombines two configurations by transferring effective agent designs or coordination patterns from one to another;

- Consolidate: summarizes an evolution trace into a compact representation that is stored in memory.

Given a task query $q$, EvoMAS applies Select to retrieve initial configurations from an initial pool, executes and evaluates them, then iteratively applies Mutate and Crossover for $n$ evolution steps to produce improved offspring. The highest-scoring configuration is selected for final execution. After each task, high-reward configurations are added to the pool and evolution traces are consolidated via Consolidate and stored, enabling the system to transfer successful patterns to future queries (Figure 1(b)).

The initial pool is seeded with cooperative MAS configurations from prior work, including hierarchical and peer-to-peer structures with specialized roles for evaluation, feedback, and aggregation. Evolutionary operators applied to these cooperative seeds naturally preserve collaborative coordination patterns. While our framework does not architecturally prohibit other agent configurations, adversarial agents with conflicting goals are not considered in this work.

### 3.2. Evolutionary Generation Process

The evolutionary process synthesizes MAS configurations through iterative refinement. For each task query $q$, EvoMAS selects initial candidates from a pool, evolves them through feedback-conditioned operators, evaluates them

with a reward function, and updates the pool and memory for future tasks.

### 3.2.1. CONFIGURATION SELECTION

Let $\mathcal{C}$ denote a pool of candidate MAS configurations. Given task query $q$, EvoMAS selects $k$ initial candidates:

$$\{C_1, \ldots, C_k\} = \mathsf{Select}(q, \mathcal{C}), \tag{1}$$

where Select retrieves configurations based on task relevance using pool metadata (task annotations, domain tags, and historical performance on similar queries). The meta-model analyzes the task requirements and selects diverse parent configurations that cover complementary capabilities (e.g., one configuration strong at decomposition, another at verification). It then instantiates them into task-specific candidates by adapting agent roles and prompts to the query.

### 3.2.2. EVOLUTIONARY OPERATORS

EvoMAS generates new candidates using two operators conditioned on execution feedback. Let $\tau_q(C_i)$ denote the execution feedback for configuration $C_i$ on task $q$. We maintain an experience memory $\mathcal{E}$ that aggregates successful evolutionary patterns from past tasks. Mutation modifies exactly one component type of a single parent, such as refining agent prompts, reassigning backbone models, updating tool lists, or rewiring the communication topology. Crossover combines two parent configurations by inheriting the topology from one parent and recombining agent-level attributes (prompts, models, tools) from both.

**Mutation.** Given candidate $C_i$ and its execution feedback:

$$C_i' = \mathsf{Mutate}(C_i, \tau_q(C_i), \mathcal{E}). \tag{2}$$

The meta-model analyzes the execution trace to identify the primary performance bottleneck, then produces an updated configuration that addresses it. Each mutation is designed to modify exactly one component type per step, enabling systematic exploration of the configuration space while isolating the effect of individual changes (see Appendix C for details).

**Crossover.** Given candidates $C_i$ and $C_j$:

$$C_{ij}' = \mathsf{Crossover}(C_i, C_j, \tau_q(C_i), \tau_q(C_j), \mathcal{E}). \tag{3}$$

The meta-model receives both parent configurations along with their respective execution traces. The offspring inherits the entire communication topology from exactly one parent, preserving structural coherence; the meta-model then selects or recombines agent-level attributes (prompts, models, tools) from both parents for each position in the inherited topology (see Appendix C).

### 3.2.3. EVOLUTION AND EVALUATION

Starting from the $k$ selected initial configurations, EvoMAS executes them on task $q$ to obtain feedback, then iterates for up to $n$ evolution steps. At each step $i$:

1. Select promising configuration(s) from the current candidate set

2. Apply Mutate or Crossover to generate offspring $C_i'$.

3. Execute $C_i'$ on task $q$ and evaluate its reward

4. Add $C_i'$ to the candidate set

Configuration quality is evaluated using:

$$R(q, C) = \mathrm{Metrics}(q, C) - \beta \cdot \mathrm{Cost}(C), \tag{4}$$

where $\mathrm{Metrics}(q, C)$ is a scalar reward from an LLM-as-judge that reviews execution results and agent trajectories across multiple aspects (correctness, completeness, reasoning quality), adapting to task-appropriate evaluation dimensions. $\mathrm{Cost}(C) = T + \alpha \cdot L$ combines the total token usage $T$ and wall-clock latency $L$ (in seconds) of a single MAS execution ($\alpha{=}1000$), distinct from the meta-model's evolution overhead. $\beta \geq 0$ balances performance against efficiency.

After $n$ steps, the candidate set contains $k{+}n$ configurations. EvoMAS returns the configuration $C_q$ with highest reward $R(q, C)$.

### 3.2.4. POOL AND MEMORY UPDATES

After processing query $q$, EvoMAS updates its state to improve on future tasks.

**Pool update.** EvoMAS processes queries sequentially. After completing query $q$ at time $t$, the best configuration is added to the pool:

$$\mathcal{C}_{t+1} = \mathcal{C}_t \cup \{C_q\}, \tag{5}$$

where $\mathcal{C}_0$ is seeded with human-designed configurations from peer-reviewed work.

**Memory update.** The evolution trace $O_q = \big((o_1, C_1', r_1), \ldots, (o_n, C_n', r_n)\big)$ is consolidated into a summary via an LLM:

$$h_q = \mathsf{Consolidate}(O_q), \tag{6}$$

which distills effective evolutionary patterns (e.g., "prompt refinement for output formatting improved accuracy by 8%"). The memory stores $e_q = (q, C_q, h_q)$ and grows as $\mathcal{E}_{t+1} = \mathcal{E}_t \cup \{e_q\}$. At each evolution step, up to 3 experiences are retrieved by task-description similarity and

provided to the meta-model, enabling it to reuse successful strategies from analogous past tasks without re-discovering them from scratch.

Detailed operator prompts are provided in Appendix D.

# 4. Experiments

## 4.1. Experimental Setup

**Datasets.** We evaluate our approach on three benchmarks covering different agentic task settings. **BBEH** (Kazemi et al., 2025) is a comprehensive benchmark for evaluating multi-step reasoning and tool-augmented problem solving. **SWE-Bench** (Jimenez et al., 2023) focuses on real-world software engineering tasks that require code understanding, modification, and validation against unit tests. Specifically, we conduct experiments on SWE-Bench-Lite and SWE-Bench-Verified. **WorkBench** (Styles et al., 2024) evaluates agent performance on realistic workplace tasks involving planning, tool use, and multi-step execution. Together, these datasets enable a systematic assessment of MAS generation across reasoning, coding, and tool-calling scenarios. The details about the datasets are described in Appendix A.1.

**Metrics.** For task-specific performance, we use the standard metrics provided by each benchmark, including *resolved rate* on SWE-Bench and *accuracy* on WorkBench and BBEH. We also evaluate MAS generation *executability* using **execution rate (ER)**, defined as the proportion of generated systems that can be successfully executed on a given task without structural or runtime failures. For each task query, we run each method for three independent trials, and all reported results are averaged over these runs.

**Evaluation Protocol.** Following the sequential online learning setting described in Section 3.2.4, we process benchmark queries one at a time. For each task query, we perform evolution using the current configuration pool and memory, then add the best configuration to the pool before processing the next query. As an evolutionary method, EvoAgent is evaluated using the same sequential protocol, allowing its population to evolve across queries. Non-evolutionary baselines (single-agent methods, predefined MAS, and other MAS generation methods) are evaluated independently on each query. Cross-query accumulation is a core capability that prior methods lack, not an unfair advantage; we isolate its contribution in Appendix Table 19, showing that per-query evolution alone (without any accumulation) already outperforms all baselines.

**Baselines.** We compare EvoMAS against four categories of methods. Details are provided in Appendix A.2.

- Direct LLM Call, which directly queries an LLM with the task input and the context via a single LLM call.

- Single Agent, where a single LLM-based agent equipped with tool access performs the task; when a dataset specifies available tools such as WorkBench, we provide the same tool set, while keeping other settings (e.g., prompts and model backbones) identical to the Direct LLM call.

- Human-Designed MAS, consisting of manually engineered MAS from prior work, which also serve as initial candidates in our MAS pool. For reasoning and planning tasks in BBEH and WorkBench, this pool includes Peer Review (Xu et al., 2023), Cross-team Orchestration (Croto) (Du et al., 2025), SMoA (Li et al., 2025), Multi-Agent Debate (Du et al., 2023), and Majority Vote. For coding tasks on SWE-Bench, we additionally include MetaGPT (Hong et al., 2024) and ChatDev (Qian et al., 2024).

- Automatic MAS Generation Methods, including AutoAgents (Chen et al., 2023), EvoAgent (Yuan et al., 2025), MAS-GPT (Ye et al., 2025), and ADAS (Hu et al., 2024).

**Implementation.** We use Claude-4-Sonnet as the default meta-model for generating MAS configurations, with access to a candidate model pool including Claude-3.5-Sonnet, Qwen3-235B-A22B, and Qwen3-Coder-480B-A35B to initialize agents in MAS. For single-agent baselines, we employ CodeAgent from SmolAgent on BBEH and WorkBench, and SWE-Agent on SWE-Bench, all initialized with Claude-3.5-Sonnet. SWE-Bench experiments use 8 parallel workers with a shared configuration pool and experience memory (Appendix A.5). Additional implementation details, including the configurations of human-designed MAS and baseline comparison methods, are provided in Appendix B.

## 4.2. Main Results

We evaluate EvoMAS on five benchmarks spanning reasoning, tool-use, and software engineering tasks. Table 1 presents results across four benchmarks (BBEH, WorkBench, SWE-Bench-Lite, SWE-Bench-Verified), with all methods evaluated using three backbone models (Claude-3.5-Sonnet, Qwen3-235B, Qwen3-480B). EvoMAS with automatic LLM selection dynamically assigns models per agent role during evolution. Full per-task breakdowns are provided in Tables 8, 9, and 10 in Appendix A.6.

**Superior Performance Across Diverse Domains.** EvoMAS with automatic LLM selection substantially outperforms all baselines across benchmarks. On BBEH, EvoMAS achieves **58.7%** accuracy, surpassing the best per-LLM EvoMAS configuration (52.8% with Qwen3-235B) by 5.9 points and the best EvoAgent result (48.2% with Claude-3.5-Sonnet) by 10.5 points. On WorkBench, EvoMAS reaches

*Table 1.* Main results across all benchmarks. We report task accuracy (%) for BBEH and WorkBench, and issue resolution rate (%) for SWE-Bench-Lite and SWE-Bench-Verified. Results are shown separately for three backbone models: Claude-3.5-Sonnet (C-3.5S), Qwen3-235B (Q-235B), and Qwen3-Coder-480B (Q-480B). **Bold** and underline indicate best and second-best performance within each backbone. "–" indicates the method was not evaluated on that setting. Gray cells denote EvoMAS results with a fixed single backbone (per-LLM), while blue cells denote EvoMAS with automatic LLM selection, where the evolutionary generator dynamically assigns backbone models per agent role.

| Category | Method | BBEH (%) | | | WorkBench (%) | | | SWE-Lite (%) | | | SWE-Verified (%) | | |
|---|---|---|---|---|---|---|---|---|---|---|---|---|---|
| | | C-3.5S | Q-235B | Q-480B | C-3.5S | Q-235B | Q-480B | C-3.5S | Q-235B | Q-480B | C-3.5S | Q-235B | Q-480B |
| Direct LLM Call | - | 20.3 | 15.1 | 11.4 | 17.0 | 15.8 | 13.7 | – | – | – | – | – | – |
| Single Agent | - | 33.2 | 37.8 | 27.4 | 40.1 | 35.6 | 30.1 | 23.0 | 36.3 | 40.8 | 33.6 | 50.2 | 56.4 |
| Predefined MAS | MetaGPT (Hong et al., 2024) | – | – | – | – | – | – | 29.7 | 42.9 | 39.7 | 40.9 | 44.3 | 43.8 |
| | ChatDev (Qian et al., 2024) | – | – | – | – | – | – | 20.3 | 32.0 | 36.0 | 26.1 | 31.2 | 36.2 |
| | Peer Review (Xu et al., 2023) | 38.4 | 46.2 | 36.0 | 32.3 | 30.8 | 28.5 | 26.0 | 41.0 | 46.1 | 35.6 | 39.4 | 41.2 |
| | Croto (Du et al., 2025) | 33.6 | 41.1 | 29.7 | 35.0 | 28.3 | 25.1 | – | – | – | – | – | – |
| | SMoA (Li et al., 2025) | 37.3 | 43.9 | 34.6 | 29.9 | 30.6 | 22.3 | – | – | – | – | – | – |
| | Majority Vote | 36.4 | 44.3 | 32.2 | 39.3 | 36.4 | 33.3 | 24.1 | 38.0 | 42.8 | 30.3 | 33.8 | 36.5 |
| | Multi-Agent Debate (Du et al., 2023) | 38.2 | 45.0 | 34.4 | 33.2 | 29.2 | 25.7 | 16.3 | 25.7 | 28.9 | 28.9 | 31.5 | 33.7 |
| MAS Generation | AutoAgents (Chen et al., 2023) | 23.8 | 30.2 | 23.4 | 40.4 | 37.8 | 34.6 | 3.6 | 14.7 | 16.4 | 8.7 | 26.4 | 28.8 |
| | MAS-GPT (Ye et al., 2025) | 18.9 | 12.1 | 18.2 | 14.4 | 18.1 | 20.6 | 0.0 | 2.3 | 4.1 | 3.1 | 7.6 | 9.4 |
| | ADAS (Hu et al., 2024) | 22.7 | 32.4 | 28.6 | 39.2 | 33.1 | 31.2 | 17.6 | 27.8 | 31.2 | 29.9 | 34.6 | 39.1 |
| MAS Evolution | EvoAgent (Yuan et al., 2025) | **48.2** | 43.5 | 36.7 | 41.8 | 36.2 | 33.4 | 24.6 | 38.8 | 43.6 | 36.1 | 51.8 | 53.3 |
| | EvoMAS (per-LLM) | 44.5 | **52.8** | **43.0** | **44.5** | **39.7** | **35.0** | **33.9** | **48.2** | **57.6** | **42.7** | **54.2** | **60.4** |
| | EvoMAS (LLM-Selection) | \| ← | 58.7 | → \| | \| ← | 48.9 | → \| | \| ← | 52.9 | → \| | \| ← | 63.8 | → \| |

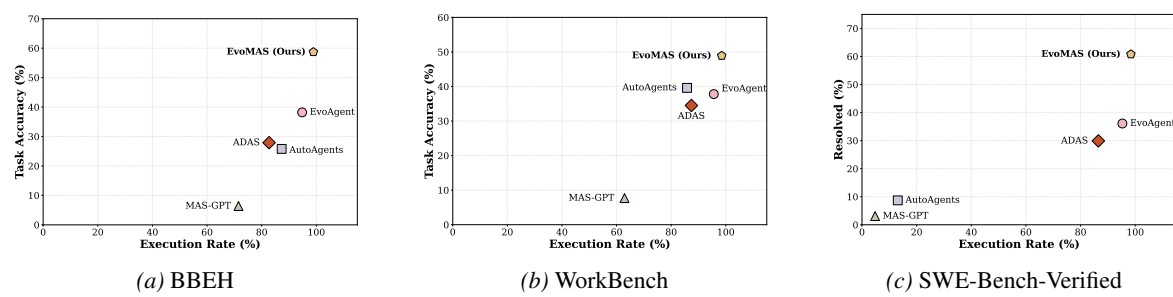

*(a)* BBEH      *(b)* WorkBench      *(c)* SWE-Bench-Verified

*Figure 2.* Trade-off between execution rate and task performance for MAS generation methods. Each point represents a method, with execution rate (%) on the x-axis and task performance (%) on the y-axis. EvoMAS achieves both high execution reliability and superior task performance across all benchmarks.

48.9%, exceeding the best per-LLM result (44.5% with Claude-3.5-Sonnet) by 4.4 points. On SWE-Bench-Verified, EvoMAS achieves **63.8%**, outperforming the best single-LLM configuration (60.4% with Qwen3-480B) by 3.4 points and surpassing Single Agent with Qwen3-480B (56.4%) by 7.4 points. Notably, MAS-GPT achieves only 3.1% on SWE-Bench-Verified with Claude-3.5-Sonnet, demonstrating that fixed code templates struggle to generalize. In contrast, our configuration-based paradigm enables flexible MAS adaptation without requiring task-specific code synthesis. Beyond these benchmarks, EvoMAS also generalizes to mathematical reasoning: on AIME 2024–2025, EvoMAS achieves 56.7% accuracy versus the best predefined MAS (Peer Review, 51.7%; Appendix Table 23).

**Robust and Reliable MAS Generation.** EvoMAS achieves consistently high execution rates: 98.9% on BBEH,

98.2% on BBEH-Mini, 98.5% on WorkBench, 96.8% on SWE-Bench-Lite, and 98.4% on SWE-Bench-Verified. This reliability stems from our configuration-based approach, where the generator produces structured configurations rather than executable code, eliminating syntax errors and runtime exceptions that plague code-generation methods. In contrast, MAS-GPT achieves only 71.5% execution rate on BBEH and 1.2% on SWE-Bench-Lite, highlighting the brittleness of code-based MAS generation.

**Outperforming Human-Designed MAS.** EvoMAS with LLM selection consistently surpasses expert-crafted MAS configurations. On BBEH, EvoMAS (58.7%) outperforms the best predefined MAS across all backbones: Peer Review with Qwen3-235B achieves 46.2%, representing a **12.5** percentage point gap. On SWE-Bench-Verified, EvoMAS achieves a **63.8%** resolution rate, substantially ex-

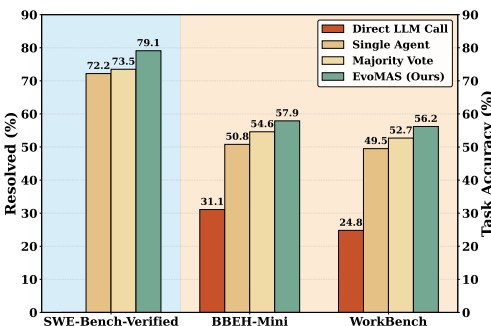

Figure 3. Results on state-of-the-art LLM (Claude-4.5-Sonnet). We compare Direct LLM Call, Single Agent, Majority Vote, and EvoMAS using Claude-4.5-Sonnet as both the MAS generator and agent backbone. EvoMAS demonstrates strong performance with the latest frontier model, achieving particularly notable results on SWE-Bench-Verified. Additional results with Claude-3.5-Sonnet and Claude-4-Sonnet are provided in Appendix Figure 5.

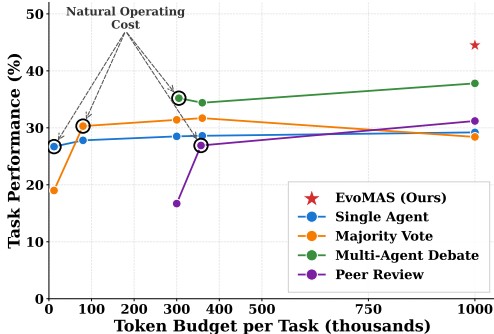

Figure 4. Scaling ability on BBEH-Mini. Each line starts at its natural operating cost (circled). EvoMAS (⋆) outperforms all baselines across budgets and continues to improve with additional compute, while other methods plateau or degrade.

ceeding the best predefined MAS results: MetaGPT reaches 40.9% (Claude-3.5-Sonnet), 44.3% (Qwen3-235B), and 43.8% (Qwen3-480B), demonstrating that evolutionary optimization discovers configurations superior to carefully designed systems across all backbone models. Predefined MAS methods exhibit variable performance across backbones and domains, as shown in Table 1. EvoMAS maintains consistently strong performance through task-adaptive evolution and dynamic model–role assignment. Detailed per-task breakdowns (Appendix A.6) show that no single model or MAS architecture universally dominates, validating our design choice to jointly evolve all configuration components, including model selection.

**Achieving improvements with scaled test-time compute.** We analyze scaling behavior under increasing token budget constraints on BBEH-Mini with Claude-3.5-Sonnet as agent backbone, where scaling is achieved by increasing the number of agents and maximum reasoning steps per agent (Figure 4). Highlighted points indicate the tokens naturally

consumed by each method without budget limits. Single-agent and predefined MAS baselines exhibit limited scaling ability. Single Agent plateaus near 29% even at 1M tokens, as additional generation alone does not improve reasoning. Majority Vote peaks around 360k tokens before degrading due to increased sampling noise. Multi-Agent Debate and Peer Review achieve moderate gains but remain constrained by fixed coordination topologies. In contrast, EvoMAS reaches 44.5% by evolving task-adaptive architectures that allocate computation more effectively across the multi-agent system. These results demonstrate that EvoMAS can make effective use of additional test-time compute to achieve better performance, while other methods cannot effectively leverage increased compute budgets. Performance gains depend not on token quantity but on how computation is structured: evolutionary optimization discovers efficient coordination patterns beyond handcrafted designs.

The budget-matched comparison (Table 6) further isolates this effect on SWE-Bench-Verified: scaling a single agent via looping to EvoMAS's token budget yields no gain, and Best-of-N with meta-model selection at 40M tokens reaches only 37.8%, while EvoMAS achieves 42.7% at 29M tokens. With Claude-4.5-Sonnet, the gap widens to 79.1% vs. 71.4% at matched 31M budgets. These results confirm that EvoMAS's gains are structural: discovering effective coordination patterns yields improvements that simple repetition or ensembling cannot achieve, even with substantially more compute.

### 4.3. Performance with State-of-the-Art LLMs

As shown in Figure 3, we evaluate EvoMAS using Claude-4.5-Sonnet, the latest frontier model, as both the MAS generator and agent backbone. EvoMAS works effectively with this state-of-the-art LLM, achieving 79.1% on SWE-Bench-Verified, slightly higher than the top performer (78.8%) on the public leaderboard at the time of writing, which used even larger LLMs, more specialized models, and agent scaffolding. EvoMAS consistently outperforms Direct LLM Call, Single Agent, and Majority Vote baselines, demonstrating that evolutionary optimization provides orthogonal benefits that complement model capability improvements. Results with Claude-3.5-Sonnet and Claude-4-Sonnet are provided in Appendix Figure 5, showing consistent performance gains across model generations: the performance gap between EvoMAS and baselines is maintained regardless of model capacity, indicating that the value of evolutionary MAS generation does not diminish as backbone models improve.

### 4.4. Analysis of EvoMAS Performance

We analyze key design choices of EvoMAS. Full ablation results are provided in Appendix A.7.

**Effect of Initial MAS Pool and Evolution Depth.** As shown in Table 2, high-quality seed configurations are critical (15.6-point gain over empty pool) and moderate depth ($T$=3) outperforms deeper evolution, suggesting diminishing returns from excessive iterations. EvoMAS consistently improves over its initialization regardless of pool quality: across the 6 non-empty pool variants ranging from curated human designs to agent-generated to overly complex seeds, evolution gains range from +1.2 to +7.6pp, and agent-generated pools reach within 1.7pp of curated pools (Appendix Table 20). Per-query evolution alone provides gains without cross-query accumulation (40.4% vs. 36.6% single agent; Appendix Table 19), confirming that the evolutionary framework is effective even in zero-shot settings where no experience transfer is possible.

*Table 2.* Effect of pool composition and evolution depth.

| Configuration | BBEH-Mini | WorkBench |
|---|---|---|
| Empty pool, $T$=5 | 33.5 | – |
| Default pool, $T$=1 | 44.4 | 45.4 |
| Default pool, $T$=3 | **49.1** | **48.9** |
| Default pool, $T$=5 | 47.6 | 48.3 |

**Backbone Model Selection.** EvoMAS supports heterogeneous model assignment where the generator dynamically assigns models per agent role. As shown in Figure 7, model size alone does not determine effectiveness; for example, Qwen3-235B outperforms the larger Qwen3-480B on BBEH (52.8% vs. 43.0%). Self-selective assignment achieves the best result at 58.7% (+5.9% over the best single-model variant), demonstrating that evolutionary discovery of model-role assignments substantially improves performance.

**Meta-Model Sensitivity.** Table 3 evaluates different meta-models on SWE-Bench-Verified with a fixed agent backbone pool. Even the weakest meta-model substantially outperforms all baselines, confirming that gains come from the evolutionary framework rather than a specific meta-model choice.

*Table 3.* Meta-model sensitivity on SWE-Bench-Verified (%).

| Meta-Model | SWE-Bench-Verified |
|---|---|
| Qwen3-235B | 58.2 |
| Claude-3.5-Sonnet | 56.4 |
| Claude-4-Sonnet (default) | 63.8 |
| Claude-4.5-Sonnet | 67.4 |
| Best non-EvoMAS baseline | 50.2 |

**Reward Signal and Hyperparameter Sensitivity.** Table 4 shows that oracle rewards provide only modest gains

over LLM-as-a-judge, confirming effective reward estimation without ground-truth labels. The judge is also robust to model choice: replacing Claude-4-Sonnet with the open-weight Qwen3-235B yields 90.4% agreement with ground truth and drops final accuracy by only 1.8pp (Appendix Table 18), confirming that evolution can be guided by accessible models without sacrificing performance. EvoMAS is insensitive to the cost trade-off parameter $\beta$: accuracy varies by only 1.1pp across $10^{-6}$ to $10^{-8}$ (Appendix Table 16).

*Table 4.* LLM-as-judge vs. oracle reward signals (%).

| Reward Signal | BBEH-Mini | WorkBench | SWE-Bench |
|---|---|---|---|
| LLM-as-Judge | 44.4 | 48.9 | 63.8 |
| Oracle (GT) | 47.8 | 50.4 | 66.9 |
| $\Delta$ | +3.4 | +1.5 | +3.1 |

**Transfer Ability.** Configurations evolved on a single BBEH subtask and transferred to the full BBEH-Mini benchmark retain up to 90% of directly optimized accuracy (best source: 44.1% vs. 49.1% original; Appendix Table 11). This transferability indicates that evolution cost can be reduced by updating on a subset of tasks while benefiting others.

### 4.5. EvoMAS Evolution Behaviors

**Emergent Behaviors and Evolutionary Patterns.** Evolved configurations exhibit emergent patterns not present in the initial pool (Appendix E): *model-role affinity* (stronger reasoning models assigned to verification roles), *functional role specialization* (dedicated verifiers and decomposers emerge), and *novel topologies* outside any pool family (collapsing agent counts by ∼70%). Evolution progresses from prompt refinement to model selection, with topology modifications later (Figure 8 in Appendix A.7). Mutation and crossover exhibit natural division of labor: mutation dominates early with targeted fixes; crossover becomes effective once configurations specialize on complementary tasks.

**Evolution Statistics and Sample Ordering.** The MAS population grows rapidly then plateaus (3→22 on SWE-Bench-Verified; 3→34 on BBEH-Mini; Figure 6 in Appendix A.7), and performance is robust to query ordering (±1.9%, std=1.1% across 11 shuffles). Table 5 shows that different task types elicit different mutation emphases, validating joint component evolution.

*Table 5.* Mutation distribution by task category on BBEH.

| Task Category | Prompt | Model | Topology |
|---|---|---|---|
| Reasoning (avg.) | 50.6% | 28.1% | 21.3% |
| Capacity-sensitive | 32.7% | 44.6% | 22.7% |
| Decomposition | 35.2% | 23.1% | 41.7% |

**Computational Cost and Equal-Budget Analysis.** Table 6 shows that EvoMAS gains are structural: it outperforms Best-of-10 (40M tokens) with fewer tokens (29M). The evolutionary overhead is bounded: the meta-model accounts for only 7.7% of tokens and 10.6% of time, while MAS execution (the useful work) dominates at 55.4% of tokens and 69.8% of time (Appendix Table 14). The pool stabilizes after ~300–400 queries, after which inference reduces to selection plus a single MAS run, amortizing the initial evolution cost. On BBEH-Mini (Figure 4), baselines plateau or degrade with more compute, while EvoMAS reaches 44.5% and continues improving.

*Table 6.* Budget-matched comparison on SWE-Bench-Verified (%).

| Method | Tokens | Accuracy |
|---|---|---|
| *Claude-3.5-Sonnet* | | |
| Single Agent | 2.3M | 33.6 |
| Loop (matched budget) | 29M | 33.6 |
| Best-of-6 + selection | 24M | 37.0 |
| Best-of-10 + selection | 40M | 37.8 |
| **EvoMAS** | **29M** | **42.7** |
| *Claude-4.5-Sonnet* | | |
| Loop (matched budget) | 31M | 71.4 |
| **EvoMAS** | **31M** | **79.1** |

## 5. Related Work

### 5.1. LLM-based Multi-Agent Systems

Large language models (LLMs) have enabled multi-agent systems (MAS) in which specialized agents collaborate through decomposition, parallelism, and diverse expertise (Cemri et al., 2025; Li et al., 2024; Guo et al., 2024). AgentVerse (Chen et al., 2024b) demonstrates that coordinated expert teams outperform single-agent systems, ChatDev (Qian et al., 2024) organizes role-specialized agents into development pipelines, and MetaGPT (Hong et al., 2024) encodes Standard Operating Procedures into prompts to reduce cascading errors. Although LLM-based MAS show strong performance across software engineering (Wu et al., 2024; Li et al., 2023a; Jiang et al., 2024) and complex reasoning (Xu et al., 2023; Du et al., 2023; Liang et al., 2024; Chen et al., 2024a), their architectures remain largely manually designed and task-specific (Zhou et al., 2025; Tran et al., 2025), motivating interest in automated MAS design.

### 5.2. Automatic Multi-Agent System Generation

Recent work explores automatic MAS generation beyond human-crafted frameworks. One line formulates MAS generation as code synthesis or open-ended search: ADAS (Hu et al., 2024) argues that learned agent designs will eventually surpass manually engineered frameworks, enabling the discovery of novel roles, tools, and workflows; MAS-GPT (Ye et al., 2025) trains an LLM to generate executable multi-agent programs directly. A complementary line incorporates optimization-based methods: GPTSwarm (Zhuge et al., 2024) (differentiable graph optimization) and AutoAgents (Chen et al., 2023) (dynamic instantiation within a fixed architecture). EvoAgent (Yuan et al., 2025) applies evolutionary mutation and crossover to agent attributes, but evolves individual agents independently rather than optimizing the multi-agent system as a whole, and lacks cross-query experience transfer. MetaAgent (Zhang et al., 2025b) models MASs as finite-state machines, enabling structured control but imposing strong architectural constraints and producing a single task-specific system rather than a population of candidate MASs.

Several concurrent works address related aspects of MAS optimization. **MASS** (Zhang et al., 2025c) optimizes MAS prompts and topology in separate stages but does not jointly evolve model assignments or tool configurations, and lacks cross-query learning. **ARG-Designer** (Chen et al., 2025) formulates topology construction as autoregressive graph generation over a fixed role pool but relies on offline training with no inference-time adaptation. **AFlow** (Zhang et al., 2025a) automates optimization of static LLM workflow graphs (pure LLM calls without agent loops or tool use), making it inapplicable to agentic tasks requiring dynamic tool invocation. In contrast, EvoMAS jointly evolves all configuration components (topology, prompts, models, tools) as a unified system and accumulates cross-query experience through pool and memory updates. Empirically, EvoMAS achieves 41.5–52.0% across backbones on BBEH-Mini reasoning tasks, versus 31.1–47.2% for ARG-Designer and 36.2–46.1% for AFlow (Appendix Tables 21 and 22).

## 6. Conclusion

We presented EvoMAS, a configuration-based evolutionary framework that synthesizes multi-agent systems by evolving modular specifications (agent roles, prompts, model assignments, and communication topologies). Operating within a bounded, interpretable configuration space, EvoMAS discovers task-adaptive architectures without manual engineering while maintaining high execution reliability (>96%). Experiments across coding, reasoning, and tool-use benchmarks show that EvoMAS consistently outperforms both predefined MAS and prior generation methods, with gains that are structural rather than compute-driven. A core design principle is trading inference-time computation for stronger results; accordingly, EvoMAS targets settings where accuracy is prioritized over latency. Extending to adversarial coordination and reducing evolution cost remain promising future directions.

## Impact Statement

This work advances automated agentic system design by showing that evolutionary optimization over structured configurations enables effective system-level test-time compute scaling. By automating multi-agent system generation, it lowers barriers to specialized agent development and may accelerate AI agent adoption across domains. As evolved multi-agent systems become more capable, human oversight and monitoring of emergent coordination patterns may become increasingly important.

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

# A. Experiment

## A.1. Dataset

We evaluate EvoMAS on five benchmarks spanning reasoning, tool-use, and software engineering domains. Table 7 summarizes the dataset statistics.

*Table 7.* Dataset statistics. BBEH contains 23 diverse reasoning subtasks; BBEH-Mini is a stratified subset with 20 samples per subtask for efficient ablation studies. WorkBench covers six workplace task categories. SWE-Bench evaluates real-world software engineering on GitHub issues.

| Dataset | Samples | Subtasks | Domain | Metric |
|---|---|---|---|---|
| BBEH | 4,520 | 23 | Reasoning | Accuracy |
| BBEH-Mini | 460 | 23 | Reasoning | Accuracy |
| WorkBench | 690 | 6 | Tool-use | Accuracy |
| SWE-Bench-Lite | 300 | – | Software Eng. | Resolved |
| SWE-Bench-Verified | 500 | – | Software Eng. | Resolved |

### A.1.1. BBEH (BIG-BENCH EXTRA HARD)

BBEH (Kazemi et al., 2025) contains 23 challenging reasoning tasks that remain difficult for state-of-the-art language models. Each task tests a specific reasoning capability:

**Logical and Mathematical Reasoning.** **Boolean Expressions** requires evaluating complex nested logical expressions combining arithmetic comparisons, boolean operators, and factual statements (e.g., capital cities). Queries contain deeply nested `not`, `and`, `or` operators with 5 candidate answers where only one evaluates to true. **Multistep Arithmetic** defines novel mathematical operators (e.g., $a\&b$, $a >< b$) with conditional semantics and requires computing expressions like $A + B - C$ where each variable involves 10+ nested operations. **Dyck Languages** tests bracket matching in sequences with multiple bracket types. **Time Arithmetic** requires temporal reasoning with time zones and date calculations.

**Language Understanding.** **Disambiguation QA** presents ambiguous sentences requiring pronoun resolution. **Hyperbaton** tests detection of unusual adjective orderings in English. **Movie Recommendation** requires identifying movies from plot descriptions and cast information. **Word Sorting** asks models to alphabetically sort lists of words.

**Spatial and Structural Reasoning.** **Geometric Shapes** requires computing properties of shapes from SVG path descriptions. **Spatial Reasoning** involves tracking object positions after described movements. **Object Counting** tests counting specific objects in complex scenes. **Shuffled Objects** tracks object swaps through multiple exchanges.

### A.1.2. WORKBENCH

WorkBench (Styles et al., 2024) evaluates tool-use capabilities in realistic workplace scenarios across six domains:

- **Analytics** (120 samples): Creating visualizations from website traffic data (e.g., "Make a bar chart of total visits since November 21")

- **Calendar** (110 samples): Scheduling meetings and managing appointments across time zones

- **CRM** (80 samples): Customer relationship management operations including contact lookup and deal tracking

- **Email** (90 samples): Composing, searching, and organizing email communications

- **Project Management** (80 samples): Task assignment, deadline tracking, and project status updates

- **Multi-Domain** (210 samples): Queries requiring coordination across multiple workplace tools

**Query Format.** WorkBench queries are natural language instructions that must be translated into structured function calls. For example, "Can you make a histogram chart of engaged users since September 27?" requires generating: `analytics.create_plot.func(time_min="2023-09-27", value_to_plot="user_engaged", plot_type="histogram")`. The ground truth specifies the exact function signature and parameters.

### A.1.3. SWE-BENCH

SWE-Bench (Jimenez et al., 2023) evaluates autonomous software engineering capabilities on real GitHub issues from popular Python repositories. **SWE-Bench-Lite** contains 300 issues filtered for tractability, focusing on well-scoped bug fixes and feature requests. Each instance provides: (1) a repository snapshot, (2) a problem statement describing the issue, and (3) a ground-truth patch for evaluation. **SWE-Bench-Verified** contains 500 human-verified instances with confirmed reproducibility and clear resolution criteria, providing a more reliable evaluation signal.

**Query Format.**    Each SWE-Bench query presents a GitHub issue with repository context:

> *"Repository: astropy/astropy. Problem Statement: Modeling's 'separability_matrix' does not compute separability correctly for nested CompoundModels..."*

The agent must autonomously navigate the repository, understand the codebase, reproduce the issue, implement a fix, and verify correctness.

### A.2. Comparison Methods

**AutoAgents** (Chen et al., 2023) maps an input query to a MAS via a fixed, hand-designed LLM pipeline in which predefined meta-roles (e.g., Planner, Agent Observer, Plan Observer) collaboratively generate agent roles and an execution plan conditioned on the query. The resulting MAS is specified implicitly in natural language and instantiated through staged prompting rather than an explicit, structured configuration. **ADAS** (Hu et al., 2024) treats the input query as a specification for open-ended search in code space, where a meta-agent iteratively synthesizes and evaluates executable agentic programs that solve the query. This process directly maps queries to MAS implementations but lacks an explicit intermediate abstraction that separates MAS topology, agent roles, and execution constraints. **MAS-GPT** (Ye et al., 2025) directly maps an input query to a complete executable MAS in a single forward pass by framing MAS generation as a conditional code generation task. The model is trained on curated query–MAS pairs, enabling it to output a query-adaptive multi-agent program without iterative search at inference time. **EvoAgent** (Yuan et al., 2025) first processes the input query using an initial Single Agent framework by evolving agent roles, prompts, and skills through mutation and crossover operators. EvoAgent evolves a population of single agents and applies a fixed ensemble strategy (e.g., random, topk, all-in) to aggregate their outputs at inference time, without explicitly constructing a query-specific multi-agent system. EvoAgent primarily evolves prompt-level agent components and does not optimize model assignment, inter-agent communication topology, or tool configurations.

### A.3. Human-Defined MAS

We include seven human-designed multi-agent systems as baselines, which also serve as initial candidates in the EvoMAS pool: Majority Vote aggregates independent responses from six parallel workers via an aggregator; Multi-Agent Debate (Du et al., 2023) runs six debaters over three rounds of all-to-all communication before a moderator synthesizes the final answer; Peer Review (Xu et al., 2023) implements a create–review–revise pipeline with six agents per stage followed by aggregation; Cross-team Orchestration (Croto) (Du et al., 2025) organizes four teams across two phases with hierarchical aggregation at each level; and SMoA (Li et al., 2025) passes inputs through two layers of four role-specialized processors, each followed by a judge that selects the best response. For SWE-Bench, we additionally include MetaGPT (Hong et al., 2024), which follows a standardized operating procedure pipeline (Product Manager → Architect → Project Manager → Engineer → QA Engineer), and ChatDev (Qian et al., 2024), which implements a chat chain pipeline (CEO → CTO → Programmer → Reviewer → Tester).

### A.4. Implementation

**Agent Backends.**    For reasoning and planning benchmarks (BBEH and WorkBench), all agents, including those in human-designed MAS and EvoMAS-generated configurations, are instantiated as CodeAgent from the smolagents framework. For SWE-Bench, we support two distinct agent backends: (1) SWE-Agent (Yang et al., 2024), the full-featured software engineering agent with repository navigation, file editing, and bash execution tools, used as the single-agent baseline; (2) CodeAgent from smolagents, used for non-worker roles such as aggregators, moderators, reviewers, and management roles (Product Manager, Architect, QA Engineer) that reason over intermediate outputs rather than manipulating code directly.

*Table 8.* Complete per-task results on BBEH benchmark. We report task accuracy (%) across 23 reasoning subtasks for each backbone model. EvoMAS achieves the highest average accuracy across all backbone configurations, with particularly strong gains on complex reasoning tasks (e.g., Zebra Puzzles, Temporal Sequences).

| Backbone | Method | BoardgameQA | Boolean Expr. | Buggy Tables | Causal Understand. | Disambig. QA | Dyck Lang. | Geom. Shapes | Hyperbaton | Linguini | Movie Rec. | Multi-step Arith. | NYCC | Object Props. | Object Counting | SARC Triples | Shuffled Objects | Spatial Reason. | SportQA | Temporal Seq. | Time Arith. | Web of Lies | Word Sorting | Zebra Puzzles | Avg. |
|---|---|---|---|---|---|---|---|---|---|---|---|---|---|---|---|---|---|---|---|---|---|---|---|---|---|
| Claude-3.5-Sonnet | Direct LLM Call | 33.5 | 22.0 | 0.5 | 51.5 | 51.7 | 16.0 | 19.0 | 1.5 | 24.5 | 68.8 | 0.0 | 7.1 | 8.5 | 2.0 | 39.0 | 11.0 | 13.5 | 25.5 | 2.0 | 10.5 | 12.0 | 13.5 | 32.5 | 20.3 |
| | Single Agent | 35.5 | 36.0 | 17.3 | 48.0 | 55.8 | 20.5 | 33.5 | 15.0 | 26.2 | 54.5 | 34.0 | 24.6 | 14.6 | 40.0 | 40.0 | 38.9 | 44.0 | 26.0 | 18.3 | 69.2 | 9.0 | 23.5 | 39.0 | 33.2 |
| | Peer Review | 27.8 | 43.0 | 25.2 | 39.2 | 63.1 | 40.3 | 35.2 | 15.8 | 28.0 | 78.7 | 49.0 | 23.3 | 12.8 | 68.3 | 43.7 | 40.5 | 53.3 | 30.8 | 15.7 | 72.5 | 13.3 | 25.2 | 39.3 | 38.4 |
| | Croto | 29.7 | 38.2 | 23.3 | 37.8 | 50.9 | 34.8 | 34.0 | 14.0 | 20.7 | 61.2 | 36.2 | 25.0 | 14.3 | 50.7 | 36.5 | 36.2 | 52.3 | 26.0 | 11.2 | 69.8 | 10.7 | 22.3 | 37.5 | 33.6 |
| | SMoA | 30.1 | 39.3 | 25.8 | 38.3 | 66.1 | 33.5 | 35.8 | 18.2 | 23.3 | 74.8 | 53.5 | 26.5 | 15.0 | 69.8 | 35.3 | 37.8 | 50.2 | 24.8 | 14.8 | 68.2 | 13.8 | 23.3 | 40.7 | 37.3 |
| | Majority Vote | 34.5 | 37.5 | 21.0 | 40.5 | 58.3 | 29.5 | 37.7 | 15.0 | 21.0 | 63.0 | 46.3 | 23.6 | 16.0 | 52.0 | 38.5 | 42.0 | 60.8 | 27.5 | 17.5 | 75.3 | 11.5 | 25.5 | 43.0 | 36.4 |
| | Multi-Agent Debate | 32.8 | 41.5 | 22.7 | 41.2 | 56.7 | 38.5 | 34.2 | 16.5 | 22.3 | 76.3 | 50.5 | 25.8 | 12.3 | 71.2 | 42.3 | 44.2 | 54.2 | 25.2 | 17.3 | 70.7 | 14.2 | 26.7 | 40.5 | 38.2 |
| | EvoMAS (Ours) | 39.5 | 45.3 | 28.2 | 44.7 | 66.7 | 46.3 | 44.5 | 19.2 | 31.2 | 87.0 | 57.8 | 29.8 | 16.8 | 77.0 | 44.5 | 49.5 | 65.7 | 32.5 | 20.5 | 78.2 | 16.7 | 33.5 | 48.3 | 44.5 |
| Qwen3-235B | Direct LLM Call | 44.0 | 29.0 | 1.5 | 47.5 | 37.5 | 7.5 | 0.5 | 1.5 | 11.5 | 61.0 | 0.0 | 3.3 | 1.5 | 0.5 | 21.5 | 6.0 | 3.0 | 24.0 | 2.0 | 13.4 | 6.0 | 7.0 | 16.5 | 15.1 |
| | Single Agent | 82.0 | 58.5 | 47.7 | 53.5 | 40.8 | 20.5 | 10.0 | 17.3 | 17.6 | 50.5 | 46.8 | 13.6 | 52.8 | 22.2 | 30.5 | 4.1 | 47.0 | 26.5 | 31.6 | 67.8 | 36.5 | 30.0 | 60.7 | 37.8 |
| | Peer Review | 83.8 | 84.3 | 58.8 | 52.5 | 48.7 | 37.7 | 9.5 | 14.3 | 19.5 | 76.2 | 58.7 | 13.2 | 64.7 | 43.5 | 34.8 | 6.5 | 52.7 | 31.2 | 45.7 | 76.2 | 37.0 | 39.8 | 74.3 | 46.2 |
| | Croto | 78.3 | 76.8 | 54.0 | 49.3 | 43.1 | 30.7 | 8.3 | 13.2 | 13.7 | 63.7 | 50.2 | 16.7 | 70.3 | 31.2 | 28.5 | 3.5 | 47.5 | 23.8 | 33.7 | 70.2 | 38.2 | 34.2 | 66.8 | 41.1 |
| | SMoA | 77.7 | 75.2 | 58.2 | 51.7 | 50.8 | 29.3 | 13.3 | 16.8 | 15.3 | 68.2 | 63.2 | 17.5 | 72.0 | 40.3 | 26.7 | 4.8 | 48.3 | 23.8 | 43.2 | 68.3 | 41.3 | 35.7 | 68.2 | 43.9 |
| | Majority Vote | 83.5 | 74.5 | 51.5 | 56.0 | 47.5 | 26.0 | 7.5 | 11.0 | 17.5 | 61.0 | 57.2 | 15.0 | 76.5 | 28.5 | 31.0 | 5.5 | 54.5 | 28.5 | 50.5 | 76.9 | 40.0 | 37.5 | 81.0 | 44.3 |
| | Multi-Agent Debate | 78.0 | 81.8 | 53.7 | 55.2 | 46.1 | 34.8 | 10.8 | 19.0 | 16.2 | 71.8 | 61.5 | 19.0 | 61.2 | 42.3 | 33.7 | 6.2 | 47.2 | 24.7 | 39.3 | 72.0 | 43.2 | 41.8 | 76.5 | 45.0 |
| | EvoMAS (Ours) | 86.8 | 87.5 | 63.7 | 59.5 | 54.0 | 44.3 | 15.2 | 22.3 | 22.7 | 86.2 | 75.7 | 21.0 | 81.7 | 51.2 | 33.8 | 7.7 | 61.8 | 32.8 | 53.3 | 80.0 | 47.2 | 47.7 | 79.2 | 52.8 |
| Qwen3-480B | Direct LLM Call | 46.5 | 25.0 | 1.0 | 46.5 | 7.5 | 7.5 | 0.5 | 1.0 | 10.0 | 25.0 | 0.0 | 0.5 | 2.0 | 0.5 | 23.0 | 3.0 | 3.0 | 20.5 | 2.5 | 7.0 | 2.5 | 8.0 | 23.5 | 11.4 |
| | Single Agent | 64.5 | 45.5 | 24.9 | 28.5 | 25.0 | 13.5 | 13.6 | 6.6 | 10.6 | 24.0 | 43.5 | 9.0 | 23.6 | 15.4 | 25.0 | 8.6 | 37.7 | 26.5 | 21.2 | 60.8 | 22.5 | 33.5 | 45.5 | 27.4 |
| | Peer Review | 76.5 | 78.0 | 32.0 | 31.0 | 41.7 | 33.0 | 22.5 | 6.5 | 16.0 | 60.5 | 47.8 | 9.5 | 25.0 | 42.5 | 28.8 | 7.3 | 46.5 | 29.5 | 21.5 | 67.3 | 24.3 | 36.8 | 42.8 | 36.0 |
| | Croto | 68.0 | 61.3 | 28.5 | 28.5 | 21.7 | 17.0 | 22.5 | 5.5 | 10.0 | 36.5 | 43.3 | 14.0 | 36.7 | 16.7 | 23.5 | 4.3 | 42.2 | 24.7 | 12.8 | 62.7 | 28.8 | 32.7 | 40.2 | 29.7 |
| | SMoA | 68.0 | 60.0 | 32.8 | 30.5 | 50.8 | 15.5 | 25.5 | 10.0 | 14.5 | 49.5 | 53.8 | 16.0 | 37.5 | 36.4 | 21.2 | 6.2 | 44.5 | 25.0 | 18.2 | 66.3 | 33.7 | 34.3 | 46.2 | 34.6 |
| | Majority Vote | 73.0 | 55.0 | 26.5 | 33.0 | 32.5 | 11.0 | 24.0 | 8.5 | 11.0 | 34.0 | 40.6 | 14.5 | 40.3 | 14.5 | 25.5 | 5.0 | 58.5 | 29.0 | 24.0 | 69.7 | 32.5 | 34.0 | 47.5 | 32.2 |
| | Multi-Agent Debate | 68.5 | 71.0 | 27.0 | 34.0 | 30.0 | 28.5 | 23.5 | 11.0 | 12.5 | 52.5 | 49.4 | 14.5 | 23.1 | 38.7 | 27.0 | 7.7 | 46.7 | 26.3 | 16.2 | 65.2 | 35.5 | 38.3 | 44.2 | 34.4 |
| | EvoMAS (Ours) | 79.5 | 81.2 | 37.5 | 37.7 | 56.2 | 30.7 | 27.8 | 13.2 | 18.5 | 69.5 | 57.2 | 18.5 | 43.5 | 47.0 | 32.5 | 8.8 | 66.3 | 29.0 | 28.2 | 73.2 | 38.5 | 43.7 | 50.3 | 43.0 |

**Comparison Methods.** We perform minimal, interface-level adaptation of each baseline's original codebase to integrate them into a unified evaluation pipeline. In all cases, the core generation or evolution logic of each method remains unchanged; only the execution interface is standardized so that all methods share the same model pool, tool access, and runtime environment. **ADAS** (Hu et al., 2024) iteratively synthesizes and evaluates executable agentic programs through a meta-agent; we provide the agent execution interface code (i.e., the smolagents `CodeAgent` API and, for SWE-Bench, the SWE-Agent API) as context so that the meta-agent can generate and revise runnable multi-agent programs within our framework. **MAS-GPT** (Ye et al., 2025) generates a complete executable MAS in a single forward pass via conditional code generation; we similarly provide the agent execution interface as the generation context and condition on the task query, so that the generated code can be directly executed in our runtime. **AutoAgents** (Chen et al., 2023) uses a Planner–Observer pipeline where predefined meta-roles collaboratively generate agent role specifications and execution plans in natural language; we provide the available agent types, model pool, and tool lists as the action space so that the generated specifications can be instantiated within our runtime. **EvoAgent** (Yuan et al., 2025) evolves a population of single agents by iteratively generating expert descriptions through mutation and merging their outputs via fixed ensemble strategies (e.g., random, top-$k$, all-in); we initialize the agent population with the same single-agent baseline used by other methods and allow EvoAgent to enrich the population through its evolutionary revision operators. In all cases, the adaptation is limited to providing the execution interface; each method's core generation or evolution logic remains unchanged.

Details of the configuration design and its interpretation logic are provided in Appendix B.

## A.5. Parallel Execution on SWE-Bench

To improve computational efficiency on SWE-Bench, we employ 8 parallel workers that share a common configuration pool and experience memory. The evolutionary process remains sequential over the benchmark: queries are processed in order, and each query's best configuration is added to the shared pool before subsequent queries begin. However, parallel workers introduce a small degree of local asynchrony in task processing and evolutionary updates, as multiple workers may simultaneously evaluate different candidate configurations or apply evolutionary operators. During our intermediate experiments, we observed that this level of parallelism does not introduce performance fluctuations larger than the inherent stochasticity of agent systems, and thus has no noticeable impact on reported results.

*Table 9.* Complete per-domain results on WorkBench benchmark. We report task accuracy (%) across six workplace task categories (Analytics, Calendar, CRM, Email, Project Management, Multi-Domain) for each backbone model. EvoMAS achieves the best average accuracy, with consistent improvements across all task domains.

| Backbone | Method | Analytics | Calendar | CRM | Email | Project Management | Multi Domain | Avg. |
|---|---|---|---|---|---|---|---|---|
| Claude-3.5-Sonnet | Direct LLM Call | 34.1 | 12.9 | 6.3 | 22.5 | 7.1 | 18.9 | 17.0 |
| | Single Agent | 39.3 | 66.1 | 35.0 | 39.3 | 23.8 | 37.1 | 40.1 |
| | Peer Review | 31.8 | 60.4 | 30.9 | 23.6 | 21.7 | 25.4 | 32.3 |
| | Croto | 42.2 | 49.6 | 39.1 | 21.3 | 21.5 | 36.1 | 35.0 |
| | SMoA | 25.0 | 53.0 | 24.3 | 23.5 | 16.7 | 37.1 | 29.9 |
| | Majority Vote | 33.5 | 71.2 | 38.9 | 37.0 | 21.3 | 33.7 | 39.3 |
| | Multi-Agent Debate | 40.6 | 41.1 | 31.0 | 22.4 | 26.3 | 38.0 | 33.2 |
| | EvoMAS (Ours) | 43.3 | 73.9 | 40.3 | 38.1 | 27.9 | 43.5 | 44.5 |
| Qwen3-235B | Direct LLM Call | 32.1 | 18.1 | 5.4 | 11.0 | 11.9 | 16.5 | 15.8 |
| | Single Agent | 39.3 | 56.2 | 32.1 | 34.8 | 16.3 | 34.8 | 35.6 |
| | Peer Review | 22.6 | 57.2 | 26.3 | 26.7 | 15.4 | 36.8 | 30.8 |
| | Croto | 20.6 | 43.8 | 27.5 | 21.5 | 18.8 | 37.5 | 28.3 |
| | SMoA | 28.8 | 52.0 | 14.5 | 33.3 | 14.6 | 40.2 | 30.6 |
| | Majority Vote | 41.6 | 60.6 | 32.7 | 34.5 | 15.0 | 33.8 | 36.4 |
| | Multi-Agent Debate | 26.0 | 49.1 | 26.7 | 32.0 | 5.8 | 35.8 | 29.2 |
| | EvoMAS (Ours) | 43.6 | 63.0 | 34.8 | 35.6 | 19.6 | 41.8 | 39.7 |
| Qwen3-480B | Direct LLM Call | 31.5 | 16.8 | 2.8 | 10.3 | 6.9 | 14.1 | 13.7 |
| | Single Agent | 34.2 | 45.2 | 29.5 | 27.0 | 16.7 | 28.1 | 30.1 |
| | Peer Review | 29.4 | 48.3 | 25.0 | 22.2 | 13.8 | 32.2 | 28.5 |
| | Croto | 26.2 | 32.1 | 26.7 | 17.1 | 12.5 | 36.2 | 25.1 |
| | SMoA | 26.2 | 42.0 | 18.5 | 12.7 | 7.1 | 27.6 | 22.3 |
| | Majority Vote | 36.3 | 50.4 | 30.5 | 31.3 | 18.3 | 32.9 | 33.3 |
| | Multi-Agent Debate | 29.0 | 38.1 | 29.1 | 20.6 | 3.8 | 33.7 | 25.7 |
| | EvoMAS (Ours) | 37.8 | 51.8 | 31.8 | 32.2 | 19.2 | 37.1 | 35.0 |

## A.6. Full Results Analysis

To mitigate stochasticity in LLM outputs, we repeat each query three times under each setting and report the average performance. Tables 8 and 9 present the complete per-task breakdown of results across all backbone models. These detailed results reveal several important patterns that support the necessity of optimizing all configuration components in EvoMAS.

**Different Models Excel on Different Tasks.** Analysis of Table 8 reveals substantial variation in model performance across task categories. On arithmetic-intensive tasks (Multi-step Arithmetic, Time Arithmetic), Qwen3-235B consistently outperforms Claude-3.5-Sonnet (e.g., 75.7% vs. 57.8% on Multi-step Arithmetic for EvoMAS). Conversely, Claude-3.5-Sonnet demonstrates stronger performance on language understanding tasks such as Movie Recommendation (87.0% vs. 86.2%) and Disambiguation QA (66.7% vs. 54.0%). Qwen3-Coder-480B, despite being larger, underperforms both alternatives on most tasks, suggesting that model scale alone does not guarantee reasoning capability. These patterns validate our design choice to include model selection as an evolvable component, as no single model universally dominates and task-adaptive model assignment enables EvoMAS to leverage each model's strengths.

**Different MAS Architectures Suit Different Tasks.** Predefined MAS methods exhibit task-dependent performance variation. On BBEH, Peer Review achieves the highest baseline accuracy on structured reasoning tasks (Peer Review: 43.0% on Boolean Expressions with Claude backbone), while Multi-Agent Debate excels on tasks requiring iterative refinement (41.5% on Boolean Expressions). For WorkBench (Table 9), Majority Vote performs well on Calendar tasks (71.2%) where consensus helps, but SMoA underperforms on Project Management (16.7%) where diverse expert perspectives may introduce confusion. EvoMAS consistently achieves the highest accuracy across all domains by evolving task-appropriate topologies, demonstrating that communication structure optimization is essential.

*Table 10.* Complete results on SWE-Bench across backbone models. We report issue resolution rate (%) for SWE-Bench-Lite (300 issues) and SWE-Bench-Verified (500 issues).

| Backbone | Method | Lite | Verified |
|---|---|---|---|
| Claude-3.5-Sonnet | Single Agent | 23.0 | 33.6 |
| | MetaGPT | 29.7 | 40.9 |
| | ChatDev | 20.3 | 26.1 |
| | Peer Review | 26.0 | 35.6 |
| | Majority Vote | 24.1 | 30.3 |
| | Debate | 16.3 | 28.9 |
| | EvoMAS (Ours) | **33.9** | **42.7** |
| Qwen3-235B | Single Agent | 36.3 | 50.2 |
| | MetaGPT | 42.9 | 44.3 |
| | ChatDev | 32.0 | 31.2 |
| | Peer Review | 41.0 | 39.4 |
| | Majority Vote | 38.0 | 33.8 |
| | Debate | 25.7 | 31.5 |
| | EvoMAS (Ours) | **48.2** | **54.2** |
| Qwen3-480B | Single Agent | 40.8 | 56.4 |
| | MetaGPT | 39.7 | 43.8 |
| | ChatDev | 36.0 | 36.2 |
| | Peer Review | 46.1 | 41.2 |
| | Majority Vote | 42.8 | 36.5 |
| | Debate | 28.9 | 33.7 |
| | EvoMAS (Ours) | **57.6** | **60.4** |

*Table 11.* Transfer experiment: MAS configurations evolved on individual BBEH subtasks are transferred to the full BBEH-Mini benchmark. *Original* denotes EvoMAS directly on BBEH-Mini.

| Source Task | Setting | Acc. (%) | $\Delta_{\text{EvoMAS}}$ |
|---|---|---|---|
| BoardgameQA → BBEH-Mini | Original | 49.1 | +12.2 |
| | Transfer | 40.7 | +3.8 |
| Boolean Expr. → BBEH-Mini | Original | 49.1 | +12.2 |
| | Transfer | 43.7 | +6.8 |
| Buggy Tables → BBEH-Mini | Original | 49.1 | +12.2 |
| | Transfer | 39.8 | +2.9 |
| DisambiguationQA → BBEH-Mini | Original | 49.1 | +12.2 |
| | Transfer | 39.3 | +2.4 |
| Dyck Languages → BBEH-Mini | Original | 49.1 | +12.2 |
| | Transfer | 40.4 | +3.5 |
| Geometric Shapes → BBEH-Mini | Original | 49.1 | +12.2 |
| | Transfer | 42.6 | +5.7 |
| Hyperbaton → BBEH-Mini | Original | 49.1 | +12.2 |
| | Transfer | 41.5 | +4.6 |
| Linguini → BBEH-Mini | Original | 49.1 | +12.2 |
| | Transfer | 37.4 | +0.5 |
| Movie Rec. → BBEH-Mini | Original | 49.1 | +12.2 |
| | Transfer | 44.1 | +7.2 |

**Tool Assignment Matters.** WorkBench tasks require agents to invoke domain-specific tools (analytics functions, calendar APIs, CRM operations). As shown in Table 9, performance varies significantly across domains: Calendar tasks achieve 73.9% (EvoMAS with Claude) while Project Management reaches only 27.9%. This gap reflects the varying complexity of tool orchestration across domains. EvoMAS allows the evolutionary process to assign appropriate tool subsets to different agents, enabling specialization that fixed tool assignments cannot achieve.

### A.7. Ablation Study

**Scaling Effects of Generator Model.** Figure 3 in the main paper and Figure 5 present our generator ablation study, where we isolate the effect of the MAS generation strategy by aligning agent backbone models with their respective generators. This controlled setup ensures that performance differences stem from the generation approach rather than model heterogeneity. Across all three backbone configurations, EvoMAS consistently outperforms baseline generators, achieving 4–8 percentage point gains over the strongest baseline (Majority Vote) on SWE-Bench-Verified. The evolutionary generator maintains its advantage regardless of the underlying model capacity: from Claude-3.5-Sonnet (54.0%) to Claude-4-Sonnet (64.6%) to Claude-4.5-Sonnet (72.2%). Similar patterns hold on BBEH-Mini and WorkBench, confirming that EvoMAS's benefits arise from its generation strategy rather than relying on a specific model's capabilities. Notably, the performance gap between EvoMAS and simpler generators (Direct LLM Call, Single Agent) widens on more challenging tasks, indicating that evolutionary optimization of multi-agent configurations becomes increasingly valuable as task complexity grows.

**Effect of Initial MAS Pool.** The default initial MAS pool of EvoMAS contains Peer Review, Majority Vote, and Multi-Agent Debate for BBEH and WorkBench, with ChatDev and MetaGPT additionally included for SWE-Bench. As shown in Table 12, starting from an empty pool ($\phi$) yields only 33.5% accuracy on BBEH-Mini after five evolution steps, whereas initializing with the default pool achieves 49.1% at $T=3$, corresponding to a 15.6-point improvement. This result demonstrates that high-quality seed configurations provide effective structural building blocks that substantially accelerate evolution. However, expanding the pool with additional predefined MAS, such as Croto or SMoA, does not further improve performance and can even degrade it. For example, adding Croto results in 44.1% accuracy compared to 49.1% with the default pool at $T=3$. These results indicate that pool diversity must be balanced against configuration quality, as larger pools may introduce suboptimal patterns that dilute the evolutionary signal and reduce execution reliability.

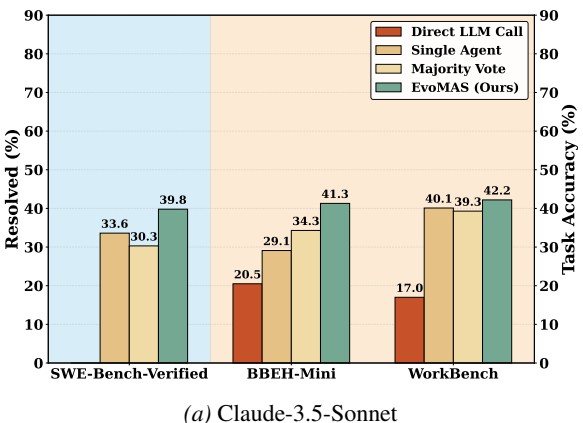

*(a)* Claude-3.5-Sonnet

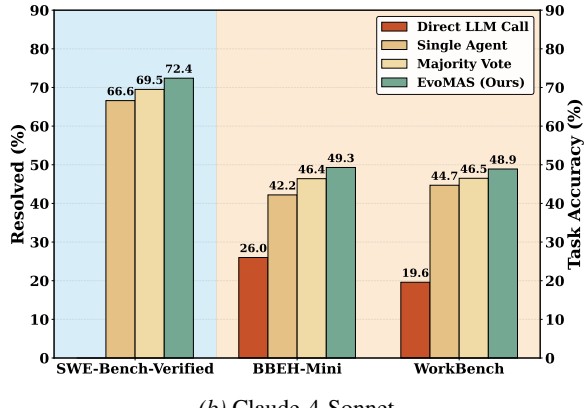

*(b)* Claude-4-Sonnet

*Figure 5.* Results with earlier Claude Sonnet models. We compare Direct LLM Call, Single Agent, Majority Vote, and EvoMAS using (a) Claude-3.5-Sonnet and (b) Claude-4-Sonnet as both the MAS generator and agent backbone. EvoMAS consistently outperforms baselines across both model generations. Performance scales with model capability, with Claude-4-Sonnet achieving substantially higher absolute performance than Claude-3.5-Sonnet across all methods.

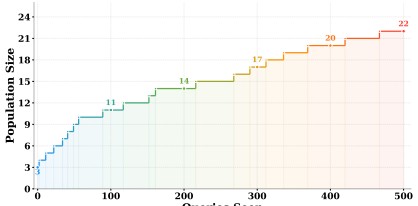

*(a)* MAS Population Growth During Evolution on SWE-Bench-Verified

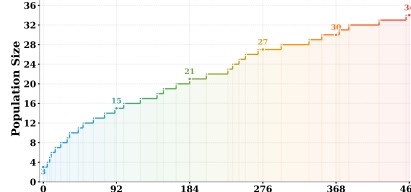

*(b)* MAS Population Growth During Evolution on BBEH-Mini

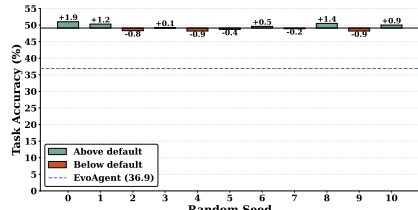

*(c)* Query Order Robustness on BBEH-Mini

*Figure 6.* Evolution behavior analysis. (a)–(b) MAS population growth during evolution on SWE-Bench-Verified and BBEH-Mini, showing how new configurations join the population pool as queries are processed. (c) Performance variation on BBEH-Mini when shuffling the query order with different random seeds, displayed as deviations from the default-order baseline (49.1%).

As shown in Figure 6(a)–(b), the population growth follows a characteristic pattern across both benchmarks: the pool expands rapidly during the early stage of evolution as the system discovers diverse, effective configurations, then gradually plateaus as the search space becomes saturated and newly generated candidates offer diminishing marginal improvements over the existing population. On SWE-Bench-Verified, the population grows from 3 to 22 over 500 queries, while on BBEH-Mini it reaches 34 over 460 queries, reflecting BBEH-Mini's broader task diversity that rewards a larger variety of MAS configurations. Figure 6(c) examines the sensitivity of EvoMAS to the order in which queries are presented during evolution. Across 11 random shuffles (seeds 0–10), the performance on BBEH-Mini remains tightly concentrated around the default-order baseline of 49.1%, with deviations ranging from $-0.9$ to $+1.9$ and a standard deviation of only 1.1%. This indicates that the evolutionary process is robust to query ordering and the performance gains reported throughout this paper are not artifacts of a particular data presentation order.

**Effective Utilization of Heterogeneous Models.** A key advantage of EvoMAS is its ability to leverage diverse backbone models within a single MAS. As shown in Figure 7, while individual backbone models exhibit varying strengths across tasks, EvoMAS with self-selective model assignment consistently achieves the best results. The evolutionary process automatically discovers optimal model assignments for different agent roles, exploiting each model's unique capabilities. For instance, EvoMAS learns to assign specialized models to particular roles (e.g., using stronger reasoning models for verification agents) without explicit guidance.

**Transfer Ability.** To evaluate the generalizability of evolved MAS configurations, we conduct a transfer experiment (Table 11) in which configurations evolved on a single BBEH subtask are directly applied to the full BBEH-Mini benchmark without further evolution. Despite being optimized for only one narrow task, the transferred configurations retain a substantial portion of the original performance, with the best source tasks (Movie Recommendation: 44.1%, Boolean

*Table 12.* Ablation study on initial MAS pool and evolution depth. EvoMAS$_{(\mathcal{C},T)}$ denotes using initial pool $\mathcal{C}$ with $T$ evolution steps. $\phi$ indicates an empty pool (single-agent only). Results show that high-quality seed configurations accelerate evolution, and $T = 3$ provides the best accuracy-efficiency trade-off.

| Method | BBEH-Mini | | WorkBench | |
|---|---|---|---|---|
| | ER. (%) | Acc. (%) | ER. (%) | Acc. (%) |
| EvoMAS$_{(\phi,1)}$ | 99.3 | 30.2 | 99.1 | 42.3 |
| EvoMAS$_{(\phi,3)}$ | **99.8** | 31.5 | **99.4** | 43.1 |
| EvoMAS$_{(\phi,5)}$ | 99.6 | 33.5 | 99.2 | 44.2 |
| EvoMAS$_{(\{\text{Peer Review},\text{Majority Vote},\text{Multi-Agent Debate}\},1)}$ | 97.8 | 45.2 | 98.1 | 46.2 |
| EvoMAS$_{(\{\text{Peer Review},\text{Majority Vote},\text{Multi-Agent Debate}\},3)}$ | 98.3 | **49.1** | 98.5 | **48.9** |
| EvoMAS$_{(\{\text{Peer Review},\text{Majority Vote},\text{Multi-Agent Debate}\},5)}$ | 98.5 | 47.6 | 98.1 | 48.3 |
| EvoMAS$_{(\{\text{Peer Review},\text{Majority Vote},\text{Multi-Agent Debate},\text{Croto}\},1)}$ | 94.3 | 43.9 | 95.2 | 44.8 |
| EvoMAS$_{(\{\text{Peer Review},\text{Majority Vote},\text{Multi-Agent Debate},\text{Croto}\},3)}$ | 95.2 | 44.1 | 95.8 | 46.7 |
| EvoMAS$_{(\{\text{Peer Review},\text{Majority Vote},\text{Multi-Agent Debate},\text{Croto}\},5)}$ | 94.8 | 45.7 | 95.4 | 45.3 |
| EvoMAS$_{(\{\text{Peer Review},\text{Majority Vote},\text{Multi-Agent Debate},\text{SMoA}\},1)}$ | 93.9 | 45.9 | 94.6 | 45.9 |
| EvoMAS$_{(\{\text{Peer Review},\text{Majority Vote},\text{Multi-Agent Debate},\text{SMoA}\},3)}$ | 94.6 | 47.2 | 95.3 | 47.4 |
| EvoMAS$_{(\{\text{Peer Review},\text{Majority Vote},\text{Multi-Agent Debate},\text{SMoA}\},5)}$ | 94.1 | 47.0 | 94.9 | 47.8 |
| EvoMAS$_{(\{\text{Peer Review},\text{Majority Vote},\text{Multi-Agent Debate},\text{Croto, SMoA}\},1)}$ | 92.6 | 42.4 | 93.4 | 44.7 |
| EvoMAS$_{(\{\text{Peer Review},\text{Majority Vote},\text{Multi-Agent Debate},\text{Croto, SMoA}\},3)}$ | 93.5 | 44.1 | 94.2 | 46.2 |
| EvoMAS$_{(\{\text{Peer Review},\text{Majority Vote},\text{Multi-Agent Debate},\text{Croto, SMoA}\},5)}$ | 93.0 | 45.2 | 93.8 | 44.9 |

*Table 13.* Ablation study on reward signal design. We compare EvoMAS using LLM-as-a-judge (default) versus task-specific pre-defined metrics for reward computation. Pre-defined metrics consistently improve performance when ground-truth labels are available, indicating that explicit reward signals reduce noise in evolution.

| Method | BBEH-Mini | WorkBench | SWE-Bench-Lite | SWE-Bench-Verified |
|---|---|---|---|---|
| | Acc. (%) | Acc. (%) | Resolved (%) | Resolved (%) |
| EvoMAS w/o Pre-defined Metrics (Default) | 44.4 | 37.6 | 14.3 | 42.3 |
| EvoMAS w/ Pre-defined Metrics | 47.8 | 39.1 | 15.8 | 45.4 |

Expressions: 43.7%) achieving close to 90% of the directly optimized accuracy (49.1%). This suggests that EvoMAS discovers generalizable MAS structural patterns, such as effective agent roles, communication topologies, and prompt strategies, rather than purely task-specific solutions. Notably, source tasks involving structured reasoning, including Boolean Expressions and Geometric Shapes, tend to transfer more effectively than tasks centered on domain-specific knowledge, such as Causal Understanding and Linguini, indicating that certain task types elicit more universally applicable MAS designs.

**Impact of Reward Signal Design.** Table 13 compares EvoMAS using LLM-as-a-judge versus task-specific pre-defined metrics for reward computation. When ground-truth labels are available, explicit task metrics consistently improve performance: accuracy increases from 44.4% to 47.8% on BBEH-Mini (+3.4%) and from 37.6% to 39.1% on WorkBench (+1.5%). On SWE-Bench-Verified, the improvement reaches +3.1% (42.3% to 45.4%). These results indicate that LLM-based judgment introduces noise during evolution, and explicit reward signals enable more effective optimization when task-specific evaluation criteria are well-defined.

**Prompt Mutations Dominate Reasoning Tasks.** Figure 8 presents the distribution of mutation operations across configuration components for all 23 BBEH subtasks. Since BBEH does not provide external tools, mutations are applied only to three component types: prompt editing, model selection, and communication topology. This distribution reveals task-dependent mutation patterns that illuminate how EvoMAS adapts configurations to different reasoning challenges. Across all BBEH tasks, prompt mutations constitute the largest fraction of successful mutations, averaging 50.6%. Tasks requiring structured reasoning, including Boolean Expressions (56.8%), Dyck Language (61.7%), Multi-step Arithmetic (57.2%), and Time Arithmetic (62.4%), exhibit particularly high prompt mutation rates. This pattern reflects the importance of

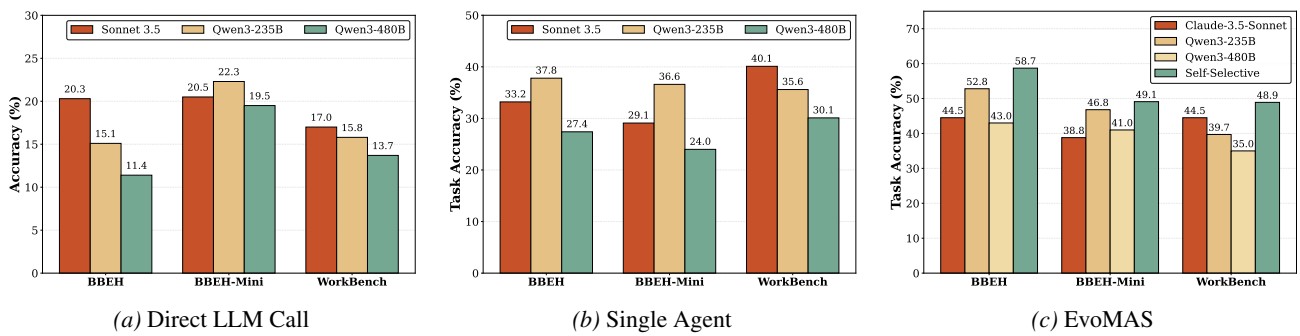

*(a) Direct LLM Call*  *(b) Single Agent*  *(c) EvoMAS*

*Figure 7.* Performance comparison across different backbone model configurations. (a) Direct LLM Call: raw model performance without agent scaffolding. (b) Single Agent: performance with tool-augmented single agent. (c) EvoMAS: our method with self-selective model assignment, where the evolutionary process dynamically assigns optimal models to each agent role. Self-Selective consistently outperforms single-backbone variants by leveraging heterogeneous model capabilities.

Chain-of-Thought prompting for mathematical and logical reasoning, as refining prompts to include step-by-step reasoning instructions substantially improves agent performance on these tasks.

**Model Selection Is Critical for Capacity-Sensitive Tasks.**   As shown in Figure 8, model ID mutations account for approximately 31.3% of mutations on average, with notable variation across tasks. Tasks with high model mutation rates include Shuffled Objects (44.6%), SARC Triples (41.8%), and NYCC (37.1%), all of which are tasks where the underlying reasoning capacity of the model directly impacts performance. Conversely, simpler classification tasks such as Dyck Language (24.1%) exhibit lower model mutation rates, as prompt improvements are sufficient to guide correct responses.

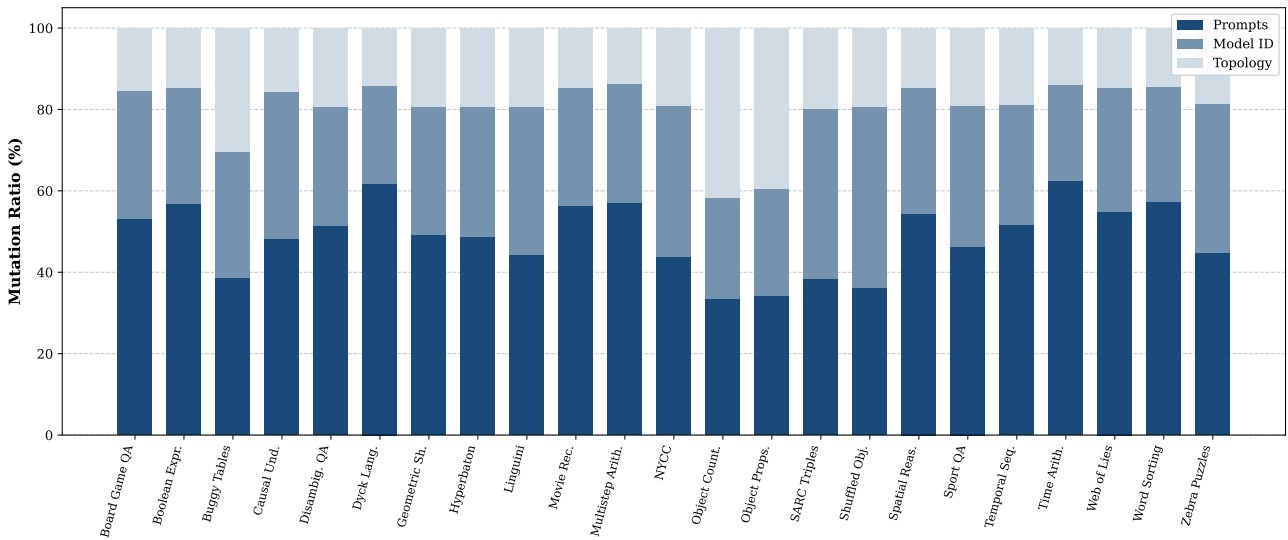

*Figure 8.* Distribution of mutation operations across configuration components for BBEH tasks. As BBEH does not provide external tools, mutations are applied only to prompt editing, communication topology, and model selection.

**Topology Mutations Address Coordination Challenges.**   Topology mutations (averaging 18.1%) are most prevalent in tasks requiring multi-step verification or decomposition. Object Counting (41.7%) and Object Properties (39.5%) exhibit the highest topology mutation rates, reflecting the benefit of decomposing counting and categorization into parallel subtasks with aggregation. Tasks with inherently sequential reasoning (Multi-step Arithmetic: 13.7%, Time Arithmetic: 13.8%) show lower topology mutation rates, as these tasks benefit more from improved prompts than from distributed coordination.

These patterns validate our design decision to evolve all three component types jointly: different tasks require different mutation emphases, and the evolutionary process automatically discovers task-appropriate optimization strategies.

## A.8. Cost Breakdown

Table 14 presents the per-task token and time breakdown for all methods across benchmarks. The key observation is that EvoMAS's computational overhead is dominated by MAS execution itself (69.8% of time, 55.4% of tokens on SWE-Bench-Verified), not by the evolutionary search: the meta-model accounts for only 10.6% of time and 7.7% of tokens. This means the overhead of evolution is modest relative to the cost of running the discovered MAS configurations. For recurring task distributions, the cost further amortizes: the pool stabilizes after ∼300–400 queries, after which EvoMAS reduces to configuration selection plus a single MAS execution (see Transfer Ability in the main paper).

*Table 14.* Computational cost breakdown per task across benchmarks. Tokens (K) and time (seconds) are averaged per query. EvoMAS phase breakdown shows that MAS execution dominates cost, while the meta-model overhead is modest.

| Method | BBEH-Mini | | WorkBench | | SWE-Bench-Verified | |
|---|---|---|---|---|---|---|
| | Tokens (K) | Time (s) | Tokens (K) | Time (s) | Tokens (K) | Time (s) |
| Single Agent | 98 | 42 | 733 | 989 | 2,300 | 1,840 |
| Majority Vote | 756 | 134 | 5,669 | 3,460 | 11,700 | 9,200 |
| Peer Review | 312 | 198 | 2,656 | 19,826 | 8,400 | 14,300 |
| Multi-Agent Debate | 486 | 274 | 8,417 | 17,993 | 9,800 | 16,500 |
| EvoMAS ($T=3$) | 952 | 385 | 7,200 | 22,400 | 28,900 | 38,600 |
| *EvoMAS Phase Breakdown (SWE-Bench-Verified):* | | | | | | |
| Meta-model | — | | — | | 2,230 (7.7%) | 4,090 (10.6%) |
| MAS Evaluation | — | | — | | 10,650 (36.9%) | 7,560 (19.6%) |
| MAS Execution | — | | — | | 16,020 (55.4%) | 26,950 (69.8%) |

## A.9. Budget-Matched Comparison

A natural question is whether EvoMAS's gains simply come from spending more compute. To answer this, we force baselines to fully utilize EvoMAS's token budget via two strategies: (1) *loop*, which re-invokes the agent until the budget is exhausted with pass@N evaluation; (2) *Best-of-N*, which runs N independent instances and selects the best via the same LLM-as-judge meta-model as EvoMAS. Table 15 shows results on SWE-Bench-Verified.

*Table 15.* Budget-matched comparison on SWE-Bench-Verified. EvoMAS achieves higher accuracy than baselines consuming equal or greater token budgets.

| Method | Tokens (M) | Resolved (%) |
|---|---|---|
| *Claude-3.5-Sonnet backbone:* | | |
| Single SWE-Agent | 2.3 | 33.6 |
| Majority Vote | 11.7 | 30.3 |
| mini-SWE-agent loop | 29 | 30.4 |
| SWE-agent loop | 29 | 33.6 |
| SWE-agent Best-of-6 | 24 | 37.0 |
| SWE-agent Best-of-10 | 40 | 37.8 |
| **EvoMAS** | **29** | **42.7** |
| *Claude-4.5-Sonnet backbone:* | | |
| Single SWE-Agent | 2.7 | 70.4 |
| mini-SWE-agent loop | 31 | 70.8 |
| SWE-agent loop | 31 | 71.4 |
| **EvoMAS** | **31** | **79.1** |

The key finding is that scaling a single agent via looping yields diminishing returns even at budgets exceeding EvoMAS's: SWE-agent loop at 29M tokens achieves only 33.6% (identical to the base 2.3M single run). Best-of-10 with meta-model selection at 40M tokens reaches 37.8%, yet EvoMAS achieves 42.7% with only 29M tokens—outperforming the most

expensive baseline with fewer tokens. Under Claude-4.5-Sonnet, EvoMAS maintains a consistent +7.7pp advantage at matched budget. These results confirm that discovering effective coordination patterns through evolution yields gains that simple repetition, ensembling, or best-of-N selection cannot achieve.

### A.10. Beta Sensitivity

The parameter $\beta$ in the reward function $R = \text{Metrics}(q, C) - \beta \cdot \text{Cost}(C)$ modulates the trade-off between task performance and token efficiency. Table 16 shows that EvoMAS is insensitive to $\beta$ across two orders of magnitude: performance varies by only 1.1pp from $10^{-6}$ to $10^{-8}$, with $10^{-6}$ and $10^{-7}$ providing the best accuracy-cost balance. At larger $\beta$ ($10^{-4}$, $10^{-5}$), the cost penalty becomes strong enough to suppress exploration, reducing both token usage and accuracy. The default $\beta = 10^{-6}$ balances Metrics$\in [0, 100]$ against Cost$\in [100K-32M]$, yielding a cost penalty of order $\sim$1–30 on the 0–100 performance scale.

*Table 16.* Sensitivity to cost-performance trade-off parameter $\beta$ on BBEH-Mini (Qwen3-235B). Performance remains stable (1.1pp variation) across two orders of magnitude ($10^{-6}$ to $10^{-8}$).

| $\beta$ | Acc. (%) | Tokens (K) |
|---|---|---|
| $10^{-4}$ | 45.4 | 709 |
| $10^{-5}$ | 46.1 | 723 |
| $10^{-6}$ (default) | 49.1 | 952 |
| $10^{-7}$ | 49.8 | 974 |
| $10^{-8}$ | 48.7 | 996 |

### A.11. Meta-Model Ablation

A potential confound is whether EvoMAS's performance is driven by the specific meta-model (Claude-4-Sonnet) rather than by the evolutionary framework itself. To isolate this, we evaluate EvoMAS with four meta-models of varying capability on SWE-Bench-Verified, keeping the agent backbone pool fixed (Claude-3.5-Sonnet, Qwen3-235B, Qwen3-Coder-480B). Table 17 shows that even the weakest meta-model (Qwen3-235B, an open-weight model) achieves 58.2%, substantially outperforming all baselines including the Qwen3-235B single agent (50.2%). Stronger meta-models improve results monotonically, but the critical observation is that a +8.0pp gain over the best baseline is achieved by the cheapest meta-model, confirming that the evolutionary framework—not the meta-model's raw capability—is the primary source of improvement.

*Table 17.* Meta-model ablation on SWE-Bench-Verified. Four meta-models drive evolution while the agent backbone pool remains fixed. All outperform the best baseline (Qwen3-235B single agent: 50.2%).

| Meta-Model | Resolved (%) |
|---|---|
| Qwen3-235B | 58.2 |
| Claude-3.5-Sonnet | 56.4 |
| Claude-4-Sonnet (default) | 63.8 |
| Claude-4.5-Sonnet | 67.4 |
| *Best baseline (Qwen3-235B single agent)* | *50.2* |

### A.12. Judge Model Ablation

EvoMAS relies on an LLM-as-judge for reward signals during evolution. A concern is whether the judge introduces biases or whether evolution is fragile to judge quality. Table 18 evaluates three judge models. Robustness comes from trace-level evaluation: structural failures (tool errors, incomplete outputs, coordination breakdowns) are detectable without ground-truth labels, making the judge's task relatively straightforward. All judges achieve >90% agreement with ground-truth oracles, and replacing Claude-4-Sonnet with the open-weight Qwen3-235B drops final accuracy by only 1.4–1.8pp. This confirms that evolution can be guided by affordable, accessible judge models without sacrificing meaningful performance.

*Table 18.* Judge model ablation. Agreement rate (%) with ground truth and downstream EvoMAS accuracy. All judges maintain >90% agreement; open-weight Qwen3-235B drops only 1.4–1.8pp.

| Judge Model | BBEH-Mini | | SWE-Bench-Verified | |
|---|---|---|---|---|
| | Agree. | Acc. | Agree. | Resolved |
| Claude-4.5-Sonnet | 95.4 | 50.4 | 97.2 | 64.4 |
| Claude-4-Sonnet (default) | 93.0 | 49.1 | 96.6 | 63.8 |
| Qwen3-235B | 90.4 | 47.3 | 92.8 | 62.4 |

## A.13. Zero-Shot / No-Accumulation Ablation

A key question is whether EvoMAS requires cross-query accumulation (pool growth and memory) to be effective, or whether per-query evolution alone provides value. This is practically important: in a zero-shot setting where only a single task from a new domain is given, no accumulation is possible. Table 19 isolates contributions by disabling pool updates and/or memory.

Even without any accumulation (no pool updates, no memory), EvoMAS achieves 40.4%—already outperforming both the Single Agent (36.6%) and Peer Review (39.8%) baselines. This confirms that per-query evolutionary search alone discovers better configurations than static human-designed MAS. Each accumulation component provides additive gains: memory alone adds +3.3pp (cross-query pattern transfer), pool alone adds +2.2pp (configuration reuse), and combining both yields the full 49.1%. Cross-query accumulation is a capability advantage—precisely what prior MAS methods lack—not an unfair comparison artifact.

*Table 19.* Ablation of accumulation components on BBEH-Mini (Qwen3-235B). Per-query evolution alone improves over baselines; each accumulation component adds further gains.

| Method | Acc. (%) |
|---|---|
| Single Agent baseline | 36.6 |
| Peer Review baseline | 39.8 |
| EvoMAS (no pool, no memory) | 40.4 |
| EvoMAS (no pool, with memory) | 43.7 |
| EvoMAS (with pool, no memory) | 42.6 |
| EvoMAS (full) | 49.1 |

## A.14. Pool Initialization Robustness

A useful iterative method should reliably improve upon its initialization regardless of starting quality. Table 20 evaluates EvoMAS across 7 pool initialization strategies of varying quality, from curated human-designed pools to degenerate configurations.

*Table 20.* Pool initialization robustness on BBEH-Mini (Qwen3-235B). "Best-in-pool" shows the best configuration's accuracy without evolution; "EvoMAS" shows accuracy after evolution.

| Pool Variant | Best-in-pool | EvoMAS | Δ |
|---|---|---|---|
| Good (base pool, 3 configs) | 42.9 | 49.1 | +6.2 |
| Good variations (perturbed) | 42.2 | 49.8 | +7.6 |
| Overly complex (4 configs incl. Croto) | 42.9 | 44.1 | +1.2 |
| Weaker (3 agent-generated) | 41.4 | 45.9 | +4.5 |
| Weaker (5 agent-generated) | 42.8 | 47.4 | +4.6 |
| Semi-degenerate (single-agent seed) | 36.6 | 40.7 | +4.1 |
| Degenerate (empty pool) | 0.0 | 33.5 | +33.5 |

The main observations are: (1) EvoMAS consistently improves over every initialization, confirming robust evolutionary dynamics; (2) stronger initialization yields higher final performance, as expected for any iterative method (analogous to neural network training with better initialization); (3) even from a completely empty pool (no seed configurations), EvoMAS generates functional MAS from scratch, achieving 33.5%; (4) overly complex configurations (e.g., Croto with 6+ agents) can actually hurt evolution (+1.2pp vs. +6.2pp for simpler pools), suggesting that pool quality should balance diversity against unnecessary complexity. The pool can also be generated by an agent: with 5 agent-generated configs, EvoMAS

achieves 47.4%, only 1.7pp below the curated base pool, confirming both the feasibility of automated initialization and EvoMAS's robustness.

## A.15. ARG-Designer Comparison

ARG-Designer (Chen et al., 2025) uses autoregressive graph generation to optimize communication topology over a fixed pool of predefined agent roles. Unlike EvoMAS, it does not jointly evolve prompts, model assignments, or tools. Table 21 compares the two on BBEH-Mini with three backbone models.

The critical finding is that ARG-Designer actually *underperforms* the single agent baseline on Claude-3.5-Sonnet (31.1% vs. 33.2%), demonstrating that optimizing topology alone without adapting agent-level components is unreliable—the generated topology may introduce coordination overhead that hurts performance when agents are not properly configured for their roles. EvoMAS consistently outperforms ARG-Designer by 4.8–10.4pp across all backbones, confirming the importance of joint component evolution.

*Table 21.* Comparison with ARG-Designer on BBEH-Mini. ARG-Designer optimizes topology only over a fixed role pool; EvoMAS jointly evolves all components.

| Method | C-3.5S | Qwen3-235B | C-4.5S |
|---|---|---|---|
| Single Agent | 33.2 | 37.8 | 44.8 |
| ARG-Designer | 31.1 | 41.1 | 47.2 |
| **EvoMAS** | **41.5** | **45.4** | **52.0** |

## A.16. AFlow Comparison

AFlow (Zhang et al., 2025a) optimizes static LLM workflow graphs, i.e., sequences of prompt-response nodes without agent loops, tool use, or multi-turn interaction. This non-agentic constraint means AFlow cannot represent MAS configurations that require iterative tool calls or inter-agent communication, limiting its applicability to agentic tasks. We evaluate AFlow with $3\times$ ensemble and self-consistency (its strongest setting) on BBEH-Mini. Table 22 shows that EvoMAS outperforms AFlow by 5.3–6.1pp across backbones, confirming that optimizing within the richer MAS configuration space (with tool use, topology, and agent specialization) provides gains that static workflow optimization cannot achieve.

*Table 22.* Comparison with AFlow on BBEH-Mini. AFlow optimizes non-agentic workflows (no tool use, no agent loops). EvoMAS outperforms by 5.3–6.1pp.

| Method | C-3.5S | Qwen3-235B | C-4.5S |
|---|---|---|---|
| Single Agent | 33.2 | 37.8 | 44.8 |
| AFlow ($3\times$ ensemble + SC) | 36.2 | 39.3 | 46.1 |
| **EvoMAS** | **41.5** | **45.4** | **52.0** |

## A.17. AIME Mathematical Reasoning

To evaluate generalization beyond the paper's primary benchmarks, we test EvoMAS on AIME 2024–2025 (60 competition-level mathematics problems) using Claude-4-Sonnet as the backbone. Table 23 shows that EvoMAS achieves 56.7%, outperforming the best human-designed baseline (Peer Review, 51.7%) by +5.0pp. Analysis of the evolved configurations reveals an emergent pattern: the meta-model consistently evolves toward a *Decompose-Solve-Verify* topology, where solution generation and verification are separated into distinct agents with specialized prompts. This pattern builds upon elements from Majority Vote (parallel diverse solutions) and Peer Review (cross-validation), while adapting agent roles to support mathematically rigorous verification—a coordination structure not present in any seed configuration.

*Table 23.* Results on AIME 2024–2025 (60 problems, Claude-4-Sonnet). EvoMAS achieves 56.7%, outperforming all predefined MAS baselines. Evolved configurations predominantly adopt a Decompose-Solve-Verify topology.

| Method | Correct | Acc. (%) |
|---|---|---|
| Single Agent | 26/60 | 43.3 |
| Majority Vote | 29/60 | 48.3 |
| Peer Review | 31/60 | 51.7 |
| Multi-Agent Debate | 25/60 | 41.7 |
| **EvoMAS** | **34/60** | **56.7** |

## A.18. Method Comparison

Table 24 provides a structured comparison of EvoMAS against related MAS optimization methods across key design dimensions. EvoMAS is the only method that jointly evolves all configuration components (prompts, models, tools, topology) within a unified structured space, while also supporting cross-query learning (pool growth and memory transfer) and inference-time adaptation (per-query evolution). Other methods optimize subsets of these dimensions: MASS optimizes prompts and topology but in separate stages without joint co-evolution; ARG-Designer optimizes topology only over fixed roles; EvoAgent evolves agent populations but without structured configuration or cross-query transfer; AFlow optimizes static workflow graphs without agent capabilities.

*Table 24.* Structured comparison of MAS optimization methods. EvoMAS is unique in jointly evolving all configuration components with cross-query learning and inference-time adaptation.

| Dimension | EvoMAS | MASS | ARG-Designer | EvoAgent | AFlow |
|---|---|---|---|---|---|
| Optimization target | All components | Prompts + topology | Topology only | Agent attributes | LLM workflows |
| Optimization space | Structured configs | Staged | Fixed roles | Agent population | Static graphs |
| Joint evolution | ✓ | ✗ | ✗ | Partial | ✗ |
| Cross-query learning | ✓ | ✗ | ✗ | ✗ | ✗ |
| Inference-time adaptation | ✓ | ✗ | ✗ | ✓ | ✗ |

## B. Configuration

### B.1. MAS Configuration Design

EvoMAS represents MAS designs using a structured, declarative configuration format, implemented here as a YAML specification. A configuration is decomposed into three semantically distinct components: *system metadata*, *agent specifications*, and *communication topology*, together with auxiliary execution directives.

**System Metadata.**   The metadata section provides a high-level description of the MAS, including its name, intended functionality, and optional references to previously successful tasks. These fields are not executed by the runtime. Instead, they provide semantic grounding for the MAS generator, supporting guided generation and adaptation across related tasks without affecting execution correctness.

**Agent Specifications.**   The agent section defines the agent set. We denote the agent-related component of configuration, which specifies the agent set, including each agent's backbone model, accessible tools, and internal policy. The generator can reliably create, modify, or reuse individual agents through localized edits, enabling efficient exploration of heterogeneous agent compositions.

**Topology.**   System-level coordination is specified through an explicit adjacency list. Edges define permissible information flow between agents, yielding a directed graph.

**Compilation.**   Given a configuration, the interpreter instantiates agents according to their specifications, and constructs the communication graph defined by the topology. All control flow, agent behavior, and coordination logic are determined at runtime from the configuration, without explicit MAS code generation.

## B.2. Configuration Schema

```yaml
name: <string>
description: <string>
successful_tasks:
- q: <string>
  notes: <string>

agents:
  <agent_id>:
    role: <enum: processor | judge | moderator | aggregator | ...>
    agent_type: <enum: CodeAgent | ...>
    model_id: <string>
    prompt: <string | prompt_id>
    tools: [<tool_id>, ...]
    max_tokens: <int>
    backend: <backend_type>
  <agent_id>:
    ...

topology:
  reports_to:
    <agent_id>: [<agent_id>, ...]
    ...

execution:
  parallel_workers: <bool>
  timeout: <int>
  max_retries: <int>
```

*Figure 9.* Schematic YAML template for EvoMAS configurations. Concrete configurations instantiate this template with task-specific agents, models, and coordination patterns.

### B.2.1. BBEH & WORKBENCH

For reasoning and tool-use benchmarks (BBEH, WorkBench), we implement the runtime based on HuggingFace's smolagents framework. The `agent_type` field specifies the agent implementation.

For WorkBench tasks, agents are configured with domain-specific tool lists (e.g., `workbench_analytics`, `workbench_calendar`) that provide access to workplace APIs. The smolagents runtime handles tool registration, execution sandboxing, and output parsing transparently.

### B.2.2. SWE-BENCH

For software engineering tasks on SWE-Bench, we adapt the official SWE-Agent (Yang et al., 2024) configuration and runtime into EvoMAS's unified schema. The `agent_type` field for SWE-Bench supports `SWE-Agent` and `CodeAgent` (enhanced with code generation capabilities). SWE-Bench configurations typically feature longer timeouts (`timeout: 1200s`) for repository exploration and instance-specific prompts that inject repository context.

### B.2.3. BACKEND ABSTRACTION

The backend can be specified at two levels: (1) the top-level `backend` field sets the default for all agents, and (2) per-agent `backend` fields override the default. This enables heterogeneous MAS where, for example, worker agents use `sweagent` for code manipulation while an aggregator uses `smolagents` for patch selection.

**Extensibility to New Frameworks.** This backend abstraction demonstrates a key advantage of our configuration-based approach: **EvoMAS can easily incorporate new agent frameworks** by wrapping their execution engines as backend modules. To integrate a new framework (e.g., AutoGPT, OpenHands), developers implement a backend adapter that translates EvoMAS configurations to the framework's native format while exposing evolvable parameters through the standard schema. This design ensures that EvoMAS's evolutionary optimization can immediately benefit from advances in specialized agent frameworks without changes to the core evolution algorithms.

### B.3. Configuration Interpretation

EvoMAS configurations are interpreted by a runtime executor (`MasRuntime`) that instantiates agents, establishes communication channels, and orchestrates execution.

**Agent Instantiation.**   For each agent in the `agents` section, the runtime selects a backend runner using the following priority: (1) the agent's explicit `backend` field, (2) auto-inference from `agent_type` (e.g., `CodeAgent`→smolagents, `DefaultAgent`→mini-swe-agent, `SWEAgent`→SWE-Agent), or (3) the MAS-level `backend` default. The selected runner then creates the agent by loading the specified model via a unified provider interface (`model_id` supporting Bedrock, OpenAI, and local HuggingFace models), initializing the prompt template, registering tools from the `tools` list (e.g., WorkBench domain APIs, SWE-Bench file tools), and configuring generation parameters. All runners implement a common `BaseAgentRunner` interface that returns a standardized `AgentResult` (content, success flag, token usage metadata), enabling heterogeneous backends within a single MAS.

**Topology Realization.**   The `topology.reports_to` field defines a directed acyclic graph (DAG) of communication edges. The runtime applies Kahn's algorithm to compute a topological ordering and groups agents into dependency levels, where agents within the same level have no mutual dependencies and can execute concurrently.

**Execution Orchestration.**   The runtime executes agents level by level. For each level, if `parallel_workers` is enabled, agents run concurrently via a thread pool; otherwise they execute sequentially. A shared `Context` object accumulates agent outputs: each agent's result is stored in `context.reports[agent_id]`, and downstream agents receive the full context of all previously executed agents appended to the task prompt. The final output is taken from the last agent in the topological order (typically an aggregator). Token usage metadata is aggregated across all agents for cost tracking.

## C. Evolutionary Operator Implementation

This section details the implementation of EvoMAS's evolutionary operators. EvoMAS represents each multi-agent system as a structured configuration specifying agent roles, system prompts, backbone model assignments, tool lists, and a communication topology (the directed graph of inter-agent message passing). A meta-model, an LLM that operates on these configurations, drives the evolutionary process. Both mutation and crossover are realized as single LLM calls to the meta-model, with structured prompts that enforce specific constraints on the output configuration space.

### C.1. Mutation Operator

The mutation operator receives four inputs: (1) the current MAS configuration as a full YAML specification, (2) execution logs containing accuracy, errors, and token usage from the MAS execution on the target query, (3) a formatted observation string summarizing current performance metrics, and (4) memory context consisting of up to 3 recent experiences retrieved by task similarity.

The meta-model first performs *root-cause analysis* on the execution trace to identify the single most impactful failure mode. It then selects exactly one component type to modify:

- **Prompts**: Refine system prompts across one or more agents (e.g., add output formatting instructions, clarify task decomposition)

- **Model IDs**: Reassign backbone models to agents (e.g., upgrade a reasoning agent from a smaller to a larger model)

- **Tools**: Modify tool lists for agents (e.g., add a code interpreter for computation tasks)

- **Topology**: Rewire the `reports_to` communication structure (e.g., add an aggregator between parallel workers)

Within the chosen component type, changes may be applied coherently across multiple agents. For example, selecting "prompts" allows refining all agents' prompts to achieve a system-wide improvement. The constraint that agents cannot be added or removed ensures the mutation explores within the structural neighborhood of the parent configuration.

The output is a complete updated YAML configuration. If the generated YAML fails validation (malformed YAML or missing required fields), the system falls back to the original configuration.

## C.2. Crossover Operator

The crossover operator receives five inputs: (1) two parent MAS configurations as full YAML specifications, (2) execution logs for both parents including accuracy metrics, (3) derived strengths/weaknesses for each parent based on relative performance, (4) the available model list, and (5) memory context.

The crossover follows a two-phase process:

1. **Topology Selection**: The meta-model must inherit the entire communication topology (`reports_to` structure) from exactly one parent. It cannot create novel topologies. This constraint preserves proven coordination patterns.

2. **Agent Recombination**: For each agent position in the inherited topology, the meta-model selects or combines agent-level attributes from either parent:

   - Take the full agent configuration from Parent 1
   - Take the full agent configuration from Parent 2
   - Create a hybrid by mixing prompts from one parent with models/tools from the other

This design mirrors biological crossover: the "chromosome" (topology) is inherited whole from one parent, while "genes" (agent configurations) can be recombined across parents.

If the offspring configuration fails validation, the system returns the better-performing parent.

## C.3. Evolution Loop

At each evolution step, the system stochastically selects mutation (probability 0.8) or crossover (0.2). For mutation, the current best configuration and its execution logs are provided. For crossover, the two highest-reward configurations from the running candidate set are used as parents. After each step, the offspring is evaluated on the target query and added to the candidate set; the best configuration (by reward) is updated, and the experience is recorded to memory for cross-query transfer.

# D. Prompt

This section presents the prompts used in EvoMAS, including the meta-model prompts for evolutionary operations and the task execution prompts for agents.

## D.1. Meta-Model Prompts

EvoMAS uses four meta-model prompts to guide the evolutionary process: **Select** (choosing parent configurations), **Generate** (adapting configurations to tasks), **Mutate** (targeted component modifications), and **Crossover** (combining parent configurations).

**Selection and Initialization Prompt.** The selection prompt serves two purposes: (1) choosing suitable parent configurations from the MAS pool for evolutionary operations, and (2) initializing task-specific MAS configurations at the beginning of evolution. When used for initialization, the meta-model selects the most promising configuration from the pool and adapts it to the target task by adjusting prompts, model assignments, or topology based on task characteristics. This dual-purpose design ensures that evolution starts from high-quality seed configurations tailored to the specific task requirements.

```
# Meta Model - Select Action

You are a meta model that selects parent MAS configurations for evolutionary synthesis.

## Task
Select the most suitable parent configurations from the MAS pool for the given task query.

## Input
**Task Query:** {{task_query}}
**Task Description:** {{task_description}}
**Available MAS Configurations:** {{pool_metadata}}
```

```
**Number of Parents to Select:** {{k}}

## Your Goal
Analyze the task requirements and select {{k}} parent configurations. Consider:
1. Task Similarity: Which configs solved similar tasks?
2. Agent Capabilities: Which structures match requirements?
3. Diversity: Select diverse configs for crossover
4. Performance History: Prioritize proven success

## Output Format
**Selected Configurations:**
```json
{
  "selected": [
    {"name": "<config_name>", "reason": "<brief reason>"}
  ]
}
```
```

*Figure 10.* Meta-model prompt for parent selection and initialization. The prompt instructs the model to analyze task requirements and select configurations based on similarity, capabilities, and diversity. For initialization, selected configurations are adapted to the target task before evolution begins.

**Mutation Prompt.**  The mutation prompt constrains the meta-model to focus on exactly one *component type* per mutation. Rather than modifying a single agent's single field, the mutation operates at the component level: if "prompts" is selected, all agents' prompts may be refined; if "model_id" is selected, model assignments across agents may be adjusted. This component-level granularity enables systematic exploration of the configuration space while maintaining coherent changes across the MAS.

```
# Meta Model - Mutate Action

You are a meta model that evolves MAS through targeted mutations.

## Task
Mutate the given MAS configuration based on **execution** observations to improve performance.

## CRITICAL CONSTRAINT
**Select EXACTLY ONE component type to mutate:**
- Prompts: Refine agent prompts across the MAS
- Model IDs: Adjust model assignments for **agents**
- Tools: Modify tool lists for relevant **agents**
- Topology: Change communication structure

**DO NOT:**
- Modify multiple component types simultaneously
- Add or remove **agents** (keep structure unchanged)

## Input
**Current MAS Configuration:** {{mas_config}}
**Execution Logs:** {{execution_logs}}
**Performance:** Accuracy: {{accuracy}}
**Observations:** {{observations}}

## Your Goal
Based on logs and performance, identify THE SINGLE
MOST IMPACTFUL component type to mutate:
1. Prompts: If **agents** misunderstand task requirements
2. Model IDs: If model capabilities are mismatched
3. Tools: If tool coverage is insufficient
4. Topology: If agent coordination is poor

## Output Format
**Root Cause Analysis:** [Identify main issue]
**Component Choice:** [prompts|model_id|tools|**topology**]
**Updated Configuration:** [Full mutated YAML]
**Expected Improvement:** [Explain expected fix]
```

*Figure 11.* Meta-model prompt for mutation operations. The prompt enforces single-component-type modifications: when a component (e.g., prompts) is selected, changes may be applied across all agents for that component, enabling coherent system-wide refinements.

**Crossover Prompt.**   The crossover prompt guides combining two parent configurations while preserving structural coherence.

```
# Meta Model – Crossover Action

You are a meta model that evolves MAS through crossover operations.

## Task
Create a new MAS by combining strengths of two parent configurations.

## CRITICAL CONSTRAINT
**Topology Inheritance:**
- MUST inherit ENTIRE topology from ONE parent
- Choose based on which has better coordination

**Agent Recombination:**
For EACH agent position, you can:
- Take config from Parent 1
- Take config from Parent 2
- Create hybrid combining both

## Input
**Parent MAS 1:** {{mas_config_1}}
- Accuracy: {{accuracy_1}}
- Strengths: {{strengths_1}}

**Parent MAS 2:** {{mas_config_2}}
- Accuracy: {{accuracy_2}}
- Strengths: {{strengths_2}}

## Your Goal
Combine best aspects of both parents:
1. Topology Selection: Choose parent's structure
2. Agent Selection: Best config per agent
3. Prompt Combination: Merge or select best prompts
4. Model Assignment: Best models per role

## Output Format
**Crossover Strategy:** [Combination approach]
**Topology Choice:** [Which parent, why]
**Agent Selection:** [Source per agent]
**Offspring Configuration:** [Full YAML]
```

*Figure 12.* Meta-model prompt for crossover operations. The prompt ensures structural integrity by requiring topology inheritance from a single parent.

### D.2. Task Execution Prompts

For baselines (Direct LLM Call, Single Agent) and MAS agent roles, we use simple, minimal prompts that present the task directly without elaborate scaffolding.

**Direct LLM Call and Single Agent.**   For Direct LLM Call and Single Agent baselines, we use a straightforward prompt that presents the question and requests an answer:

```
You are a worker agent in a multi-agent system.

Your role is to analyze the given task and provide your analysis or solution.

**IMPORTANT**: If your solution involves tool/function calls, you MUST list them at the end in a section
    labeled "FUNCTION_CALLS:" using the format: domain.function.func(param="value")

Please work on the following task:

{{task}}
```

*Figure 13.* Simple task execution prompt used for Single Agent baseline and worker agents in MAS configurations. The prompt provides minimal scaffolding to enable fair comparison across methods.

This minimal prompt design ensures that performance differences between methods arise from the MAS architecture and evolutionary optimization rather than from prompt engineering advantages. All baseline methods and EvoMAS-generated configurations use equivalent base prompts, with EvoMAS evolving task-specific refinements through the mutation process.

# E. Generated MAS Examples

This section analyzes the meta-model's behavior during MAS evolution by examining the architectural decisions it makes, the novel configurations it produces, and how far evolved systems diverge from their ancestors in the initial pool. We frame the analysis from the meta-model's perspective: what patterns does it observe in execution traces, what reasoning drives its mutation and crossover decisions, and what emergent designs result from iterative trace-guided refinement?

## E.1. Emergent Role Specialization

The initial pool contains configurations where agents within each functional layer are *identical*: six uniform `worker` agents in Majority Vote, six identical `debater_initial` agents in Debate's first round, or four homogeneous `team_worker` agents in Croto. A recurring pattern in evolution is that the meta-model discovers *functional specialization* by inventing roles absent from the initial pool through diagnosing specific failure modes in execution traces.

**Observation 1: Correlated Failures Reveal Missing Verification.**   When the meta-model executes Majority Vote on a boolean expression evaluation task, the trace reveals that 5 of 6 workers produce the same wrong answer due to an identical parenthesization error in translating deeply nested `not not not` chains to Python. The critical insight from the trace is that the aggregator, despite receiving a 5:1 vote for the wrong answer, recovers the correct result by *independently re-executing* the candidate expressions in its own code sandbox.

This trace pattern leads the meta-model to a non-obvious conclusion: the aggregator's incidental verification behavior is more valuable than the five redundant worker computations. The meta-model's response is not to tune the workers' prompts or temperatures (which would be the naïve mutation) but to *restructure the system around verification as a first-class role* by reducing from six identical workers to two diverse solvers (with different temperatures to break correlation) plus a dedicated verifier. The verifier role, which exists in no pool configuration, emerges from the meta-model recognizing that the aggregator accidentally performed the function that actually mattered.

**Observation 2: The "Decomposer" Role for Sequential Bottlenecks.**   On custom-operator arithmetic tasks requiring $\sim 60$ nested operator evaluations across three variables ($A$, $B$, $C$), the meta-model observes in the trace that every worker correctly implements the operator definitions as Python functions but accumulates errors in the later stages of the sequential evaluation chain. With `max_tokens=4096`, the code generation exhausts its budget midway through the computation.

The meta-model invents a **decomposer** role, an agent whose sole function is to parse the task's algebraic structure and partition it into independent sub-computations. This agent produces no solution itself; it generates a work plan that assigns separate variables ($A$, $B$, $C$) to dedicated **sub-solvers**, each operating with a larger token budget (`max_tokens=8192`) on a shorter computation. A final **combiner** agent performs only the trivial arithmetic $A + B - C$. This decomposer-solver-combiner pipeline is structurally novel: it is not a debate (no iterative argument), not a vote (no redundancy), and not a review (no critique-revision cycle), but a *task-aware DAG* whose shape is determined by the algebraic structure of the problem itself.

**Observation 3: Pipeline Collapse under Conflicting Instructions.**   For SWE-Bench, the pool includes MetaGPT-style pipelines (Product Manager → Architect → Project Manager → Engineer → QA Engineer) and ChatDev-style pipelines (CEO → CTO → Programmer → Reviewer → Tester). When the meta-model examines these traces, it identifies that upstream planning agents (Product Manager, Architect, Project Manager) each produce lengthy specifications, but only the Engineer agent actually reads files and writes patches. More critically, the trace reveals that upstream agents sometimes generate *contradictory* instructions (e.g., the Architect specifies a refactoring strategy that conflicts with the Project Manager's scope constraint), and the Engineer, confronted with conflicting upstream context, produces a patch that satisfies neither.

The meta-model's response is drastic. It collapses the five-stage pipeline into a three-stage **plan** → **branch** → **select** pattern, consisting of a single Planner that produces one coherent plan, two to three Workers that independently attempt the fix, and a Judge that selects the best patch. This pattern is absent from the initial pool and represents a qualitative architectural shift:

replacing deep sequential information processing with shallow parallel execution, eliminating the stage where conflicting instructions can accumulate.

## E.2. Novel Topology Synthesis

The initial pool spans five distinct topology families, summarized in Table 25.

*Table 25.* Topology characteristics of pool configurations. Edges count directed communication links in the `reports_to` DAG.

| Config | Agents | Edges | Pattern |
|---|---|---|---|
| Majority Vote | 7 | 6 | Star |
| Debate | 19 | 78 | Layered all-to-all |
| Peer Review | 19 | 78 | Pipeline (create→review→revise) |
| Croto | 14 | 16 | Hierarchical tree |
| SMoA | 11 | 16 | Layered bottleneck |

Evolution produces topologies that fall *outside* these five families. We highlight three novel patterns.

**Sparse Debate.**  The debate topology uses all-to-all broadcasting between rounds: each of 6 Round-1 debaters reports to all 6 Round-2 debaters, yielding $6 \times 6 = 36$ edges per layer transition (78 total edges across the DAG) and high token cost per task. When the meta-model examines the trace, it observes that Round-2 agents receive 6 near-identical messages (since all Round-1 agents see the same input query) and produce responses that are heavily influenced by the majority opinion rather than by independent reasoning. The topology mutation replaces all-to-all broadcasting with *paired communication*: agents are grouped into pairs that exchange arguments locally, and only pair-level summaries are forwarded to the next round. This reduces inter-round edges from 36 to 6 (3 intra-pair exchanges + 3 summary edges), cutting token cost by $\sim$60% while preserving the iterative refinement mechanism. The resulting topology, a paired-sparse multi-round structure, does not correspond to any existing MAS design pattern in the literature.

**Task-Decomposed DAG.**  As described in Section E.1, the decomposer-solver-combiner pattern produces a topology whose *shape depends on the task's structure*. For a three-variable arithmetic problem, the DAG has fan-out 3 from the decomposer, parallel execution of sub-solvers, and fan-in 3 at the combiner. For a table-parsing task, the decomposer might split the problem into "parse the table" and "compute the query," yielding fan-out 2. This is fundamentally different from pool topologies, which have fixed shapes regardless of the task.

**Asymmetric Depth via Crossover.**  Crossover can produce topologies with *asymmetric depth*, in which different processing paths have different lengths, that exist in neither parent. Consider a crossover between a post-mutation Majority Vote descendant (3 agents: 2 solvers → 1 verifier, depth 2) and a Debate descendant (7 agents: 3 Round-1 → 3 Round-2 → 1 moderator, depth 3). The meta-model inherits the Debate parent's topology skeleton but, when recombining agents, assigns solver-style prompts to one Round-1 position and leaves the others as debaters. The result is a hybrid where one "fast path" (solver → moderator, effective depth 2) coexists with a "deliberative path" (debaters → Round-2 refinement → moderator, depth 3) within the same DAG. This asymmetric design, which allows easy sub-problems to resolve quickly while harder sub-problems undergo additional refinement, is an architectural motif that cannot be expressed within any single pool topology family.

## E.3. Mutation vs. Crossover: The Meta-Model's Decision Boundary

The meta-model alternates between mutation (local refinement of one configuration) and crossover (recombination of two configurations) at each evolution step. We analyze what drives this decision.

**Mutation Dominates Early Evolution.**  In the first evolution step, the meta-model almost always chooses mutation. The reason is structural: the pool configurations are "complete" systems with internally consistent designs (e.g., Debate's all-to-all topology matches its debater prompts), and execution traces typically reveal a single dominant failure mode, such as an insufficient token budget, a prompt missing a key instruction, or a topology bottleneck. Mutation addresses this with a targeted fix: increase `max_tokens` for one agent, refine one agent's prompt to emphasize code-based computation, or restructure the topology to reduce redundant edges.

The mutation constraint of modifying *exactly one* component type per step forces the meta-model to identify the single most impactful change and prevents redesigning the entire system at once. This constraint is critical: without it, the meta-model tends to propose configurations that are internally inconsistent (e.g., changing prompts that no longer match the topology).

**Crossover Becomes Valuable When Configurations Specialize.**    After 1–2 mutation steps, pool configurations begin to specialize: one descendant excels on computation-heavy tasks (due to increased token budgets and solver prompts) while another excels on knowledge tasks (due to debate-style cross-checking). At this point, crossover becomes the meta-model's preferred operator, because it can combine complementary strengths.

The crossover decision involves two sub-decisions:

1. **Topology selection**: The meta-model must commit to one parent's entire `reports_to` structure. This is an architectural decision, such as choosing "star" vs. "layered" vs. "pipeline", that determines the offspring's interaction pattern. The meta-model selects the topology that performed better on the *current query's task type*, not on aggregate accuracy.

2. **Agent recombination**: For each agent position in the inherited topology, the meta-model chooses the agent configuration from Parent 1, Parent 2, or a hybrid. The trace informs this: if Parent 1's solver prompt produced cleaner intermediate reasoning but Parent 2's aggregator prompt synthesized results more reliably, the meta-model inherits each from its respective parent.

**When Crossover Fails.**    Crossover is ineffective when both parents share the same fundamental limitation (typically, the same underlying model). On sarcasm detection tasks, the meta-model observes that *every* configuration in the pool produces the same wrong answer (e.g., labeling verbal irony as sarcasm), regardless of topology or prompt. In these cases, no crossover can improve performance because the error is not architectural but is a systematic bias in the base LLM. The meta-model's trace shows unanimous agent agreement with internally coherent (but incorrect) justifications, producing no actionable signal for either mutation or crossover. This is a boundary where evolutionary MAS synthesis reaches its fundamental limit: it can optimize the *orchestration* of model calls but cannot correct biases that are uniform across all models in the pool.

### E.4. Structural Divergence from Pool Ancestors

A natural question is how far evolved configurations diverge from their pool ancestors across evolution steps. We characterize divergence along four axes.

**Agent Count Reduction.**    The most consistent trend across evolution runs is *agent count compression*. Pool configurations range from 7 agents (Majority Vote) to 19 agents (Debate, Peer Review), but evolved configurations converge to 3–5 agents. This is not a hard-coded preference, as the meta-model is free to add agents during generation, but instead emerges from trace analysis. The meta-model consistently observes that redundant agents, such as identical workers in Majority Vote or planning-stage agents in SWE-Bench pipelines, consume tokens without contributing to correctness. After 3 evolution steps, the typical evolved configuration has ∼70% fewer agents than its pool ancestor.

**Edge Density Collapse.**    Topology divergence is even more dramatic. Debate's all-to-all connectivity (78 directed edges) is the most expensive pattern in the pool. Evolution reduces this to sparse or paired topologies with 4–8 edges, corresponding to a ∼95% reduction, while preserving the iterative refinement property through targeted rather than broadcast communication. The evolved topologies resemble neither the dense debate graph nor the simple star of Majority Vote, but occupy a previously unexplored region of the topology space.

**Role Diversity Increase.**    While pool configurations use 2–3 role types (e.g., `worker` + `aggregator`, or `debater` + `moderator`), evolved configurations typically use 3–4 distinct roles. These roles emerge from trace analysis rather than from templates: `decomposer`, `solver`, `verifier`, `combiner`, `devil's advocate`, `planner`, `judge`. The proliferation of roles reflects the meta-model's tendency to assign *exactly one function* per agent, a design principle it discovers rather than inherits.

**Prompt Divergence.**    After 3 evolution steps, the evolved configuration's prompts share minimal overlap with any pool configuration's prompts. Initial mutations tend to be additive (appending instructions like "always verify your computation

by re-executing the code"), but crossover and subsequent mutations produce prompts that are qualitatively different from both parents. By the third step, the meta-model has effectively authored novel role descriptions that encode task-specific strategies gleaned from accumulated trace observations across prior evolution steps.

**Summary.**    Combining these axes, a typical 3-step evolution trajectory transforms a 19-agent, 78-edge Debate configuration into a 4-agent, 5-edge solve-verify-combine pipeline (a system that is architecturally unrecognizable from its ancestor). The evolved design is not a member of any named MAS pattern (Majority Vote, Debate, Peer Review, etc.) but a bespoke architecture that the meta-model synthesized through iterative trace analysis. This capacity to exit the initial design space and discover genuinely novel multi-agent architectures is the central capability that distinguishes evolutionary MAS synthesis from template-based or hand-designed approaches.

