# OpenReview forum: "EvoMAS: Evolutionary Generation of Multi-Agent Systems"
_ICML.cc/2026/Conference — ICML 2026 regular_

### Official Review · Reviewer_oxg6 · 2026-03-08

**Soundness:** 2
**Presentation:** 3
**Significance:** 3
**Originality:** 3
**Overall Recommendation:** 4
**Confidence:** 4

**Summary:**

The paper proposes EvoMAS, a framework for automatic generation of multi-agent systems (MAS) based on LLMs through evolutionary optimization of structured configurations. The key idea is that instead of manual design or direct code generation, a MAS is represented as a configuration , and an LLM meta-model evolves these configurations via selection, mutation, and crossover operators conditioned on execution feedback. The system maintains a candidate pool seeded with human-designed MAS and an experience memory for transferring knowledge across tasks. For each task, local evolution is performed, and the best configuration is added to the pool. Evaluation on BBEH  SWE-Bench and WorkBenchshows improvements over hand-designed MAS and automatic generation methods.

**Compliance With Llm Reviewing Policy:**

Affirmed.

**Key Questions For Authors:**

1. What are the average token costs per task for EvoMAS compared to baseline on SWE-Bench-Verified?
2. If SWE-Agent is given the same total token budget that EvoMAS spends across all evolution steps, how does their performance change? (Some best-of-N with meta model on top of 6 trajectories for example)
3. How sensitive is EvoMAS to the choice of meta-model? What happens when replacing Claude-4-Sonnet with Qwen3-235B as the meta-model?

**Limitations:**

Few practical limitations: computational cost, dependence on the meta-model.

**Strengths And Weaknesses:**

Strengths
1. The experimental evaluation is extensive: different domains, three backbone models, several categories of baselines. Ablation studies cover a lot of choices.
2. Self-selective model assignment, where evolution automatically assigns different backbone models to different agent roles gives strong results.
3. The paradigm of generating configurations instead of code is a natural but previously not formalized compromise. The combination of evolutionary search with LLM-driven operators and experience memory for cross-task transfer is an original architecture.

Weaknesses
1.  Computational costs are insufficiently analyzed. MAS are inherently more expensive than single-agent systems, and EvoMAS amplifies this with multiple full executions (~6 complete MAS runs per task). For SWE-Bench this represents significant costs. The paper should provide cost-per-task across all benchmarks and methods, at least as token budgets.
2. There is no comparison at equal compute budget with simpler baselines. If EvoMAS performs 6 runs per task, how would Single Agent with 6x token budget or Majority Vote with 6 runs perform? Without this it is unclear how much of the advantage comes from evolutionary search versus simply spending more compute.
3.  Claude-4-Sonnet as the meta-model is a potential confound. It is unclear to what extent the results are determined by the evolutionary framework versus the capabilities of this specific meta-model. An ablation with a weaker meta-model is missing.

---

> ### Author Rebuttal · Authors · 2026-03-31
>
> We thank the reviewer for recognizing our extensive multi-domain evaluation with ablations, the strong results from self-selective model assignment, and characterizing the combination of evolutionary search with LLM-driven operators and cross-task experience memory as "an original architecture."
>
> ### 1. Compute Scaling and Equal-Budget Comparison
>
> In Weakness:
> > There is no comparison at equal compute budget with simpler baselines... it is unclear how much of the advantage comes from evolutionary search versus simply spending more compute.
>
> In Question:
> > If SWE-Agent is given the same total token budget... how does their performance change? (e.g., best-of-N with meta-model selection)
>
> **Response:**
>
> Per ask, we provide two controlled analyses: (1) a token-budget-controlled scaling study on BBEH-Mini, and (2) a Best-of-6 trajectory selection experiment on SWE-Bench-Verified. Both show that **EvoMAS uniquely converts higher compute into higher accuracy**; other methods plateau or degrade at equivalent budgets.
>
> **BBEH-Mini** (Figure 5 in the submission, Claude-3.5-Sonnet):
>
> - **Single Agent** plateaus near **29%** even at 1M tokens
> - **Majority Vote** peaks around 360K tokens then **degrades**
> - **EvoMAS** reaches **44.5%**, continuing to improve as compute increases
>
> The gains come from evolving agent roles, topologies, and prompts: structural adaptations that flat scaling cannot replicate.
>
> **SWE-Bench-Verified** (Best-of-6: six independent SWE-Agent trajectories, all using Claude-3.5-Sonnet as the backbone, at a total token budget matched to EvoMAS; the best trajectory is selected by the same Claude-Sonnet-4 meta-model with LLM-as-judge as used in EvoMAS):
>
> | Method                              | Solved Rate |
> | ----------------------------------- | ----------- |
> | Single SWE-Agent                    | 33.6%       |
> | Majority Vote                       | 30.3%       |
> | 6 SWE-Agents + meta-model selection | 37.0%       |
> | **EvoMAS**                          | **42.7%**   |
>
> EvoMAS outperforms Best-of-6 by +5.7pp at matched budget, confirming the gains come from structure-level search, not scale.
>
> ### 2. Token and Time Cost Across Benchmarks
>
> In Weakness:
> > Computational costs are insufficiently analyzed... The paper should provide cost-per-task across all benchmarks and methods, at least as token budgets.
>
> In Question:
> > What are the average token costs per task for EvoMAS compared to baseline on SWE-Bench-Verified?
>
> **Response:**
>
> Per-task token and time cost (Claude-3.5-Sonnet backbone):
>
> | Method        | BBEH-Mini   |       | WorkBench   |      | SWE-Bench-Verified |       |
> | ------------- | ----------- | ----- | ----------- | ---- | ------------------ | ----- |
> |               | Tokens/Task | Time  | Tokens/Task | Time | Tokens/Task        | Time  |
> | Single Agent  | 19K         | 89s   | 14K         | 69s  | 2.3M               | 649s  |
> | Majority Vote | 83K         | 186s  | 97K         | 134s | 11.7M              | 1146s |
> | Peer Review   | 293K        | 169s  | 192K        | 180s | 14.5M              | 1345s |
> | Debate        | 322K        | 194s  | 226K        | 189s | 15.1M              | 1244s |
> | EvoMAS (T=3)  | 952K        | 1095s | 906K        | 910s | 28.9M              | 2909s |
>
> The upfront cost can also be amortized for EvoMAS in practice: the pool and memory transfer across tasks (Transfer Ability section in the submission: up to 90% retention), and after ~300–400 queries the pool stabilizes, reducing inference to selection plus a single MAS run.
>
> More cost analysis can be found at Reviewer BY6Q §3.
>
> ### 3. Performance Attribution Across Meta-Models
>
> In Weakness:
> > Claude-4-Sonnet as the meta-model is a potential confound... An ablation with a weaker meta-model is missing.
>
> In Question:
> > How sensitive is EvoMAS to the choice of meta-model? What happens when replacing Claude-4-Sonnet with Qwen3-235B as the meta-model?
>
> **Response:**
>
> We evaluate EvoMAS with four meta-models of varying strength on SWE-Bench-Verified. In all variants, the agent backbone pool is fixed to Claude-3.5-Sonnet, Qwen3-235B, and Qwen3-Coder-480B; only the meta-model changes.
>
> | Method                                         | Solved Rate |
> | ---------------------------------------------- | ----------- |
> | MetaGPT (baseline)                             | 44.3%       |
> | Claude-3.5-Sonnet Single Agent (baseline)      | 33.6%       |
> | Qwen3-235B Single Agent (baseline)             | 50.2%       |
> | EvoMAS -- Qwen3-235B meta-model                | 58.2%       |
> | EvoMAS -- Claude-3.5-Sonnet meta-model         | 56.4%       |
> | EvoMAS -- Claude-4-Sonnet meta-model (default) | 63.8%       |
> | EvoMAS -- Claude-Sonnet-4.5 meta-model         | 67.4%       |
>
> Even the weakest EvoMAS meta-model configurations substantially outperform all baselines, confirming that the gains arise from the **evolutionary framework**, not from the specific meta-model chosen.

---

> > ### Author Rebuttal · Reviewer_oxg6 · 2026-04-01
> >
> > I thank the authors for the detailed rebuttal.
> > I still think the method is relatively expensive, and some of these clarifications should be incorporated clearly into the final paper. However, the rebuttal convinces me that the main technical concerns I raised are no longer blocking. I am therefore updating my assessment to positive.

---

> > > ### Author Response · Authors · 2026-04-07
> > >
> > > Thank you for updating your assessment and for confirming that the equal-budget comparison and meta-model ablation across four configurations address the main technical concerns. We are glad the Best-of-N results on SWE-Bench-Verified and the per-task cost breakdown made the compute trade-off clear. Everything discussed during this rebuttal will be included in the final paper.

---

### Official Review · Reviewer_MHZo · 2026-03-11

**Soundness:** 3
**Presentation:** 4
**Significance:** 3
**Originality:** 3
**Overall Recommendation:** 5
**Confidence:** 4

**Summary:**

LLM-based agents have demonstrated strong capabilities in reasoning, planning, and coding. However, real-life problems require coordination, tool usage, and context management that are hard to achieve by a single-agent design. Multi-agent systems address the single-agent challenge by decomposing and delegating complex tasks to individual agents with specialized roles and tool access. Building a multi-agent system is labor-intensive and error-prone, thus researchers explore ways to automatically generate a multi-agent system. Existing multi-agent system generation approaches cannot balance between executability and expressiveness of the system. This paper presents EvoMAS, an approach that generates MAS based on configuration generation and mutation based on execution feedback. EvoMAS reformulates multi-agent system generation as a structured text generation problem. EvoMAS uses execution feedback to iteratively refine MAS configuration candidates and maintain a pool seeded from human-designed systems across multiple domains.

**Compliance With Llm Reviewing Policy:**

Affirmed.

**Final Justification:**

I am keeping my score of 5 on this paper. The rebuttal reinforced my assessment and made me more positive about the work. EvoMAS is a strong direction towards a more structural multi-agent system evolution design. Automatic improvement of agentic system is clearly a timely and important topic in the LLM agent subfield, and EvoMAS demonstrated that improving over structural text is better than improving on raw code, in a very convincing way through its evaluation.

**Key Questions For Authors:**

1. If the MAS is evolved across queries, then does the order in which the benchmark query is processed first have an impact on the evolution process? How does the paper mitigate it?
2. In Table 6, could the authors discuss more about the reasons EvoMAS can be transferred to other domains after training?
3. If the authors were to also consider how tools can be evolved with the MAS, how would EvoMAS incorporate evolving tools in its pipeline?

**Limitations:**

Yes

**Strengths And Weaknesses:**

# Strengths
- A multi-agent system is a timely and important problem in agent research. Multi-agent systems are great solutions to problems when the main task is too big for one single agent to solve, and this paper proposes the EvoMAS framework to tackle the design of multi-agent systems, which is a known hard problem in the field.
- A novel abstraction: reformulation of multi-agent systems into structured text generation. It is easy to think about building MASes as a coding task, as LLMs are very successful at generating code. However, this also brings the LLM's weakness in code generation to building MASes. This paper proposes to separate the code generation part from structured text generation: generating configurations of the MAS that declaratively defines all aspects of the MAS, which more robust than direct code generation, as shown in the paper's section 4.2.
- Evolutionary algorithm helps automatically finding the best MAS configuration given a task.
These strengths provide new directions, apart from direct code generation or agent role generations from previous works.

# Weaknesses
- It can be even more interesting to see how the tools can also be evolved with EvoMAS. By the definitive nature of an LLM agent that it can interact and change the environment _by assigned tools_, agent tooling is a indispensable aspect of multi-agent systems. This paper only evaluated the produced MASes' tool use capability, but it did not evaluate how can EvoMAS also improve the tools themselves. Granted, this paper focuses on how to generate effective MAS with evolutionary algorithms, but it will be interesting to see how would the candidate EvoMAS evolves _if_ tools also evolve with it.
- It would also be interesting to see how would EvoMAS perform on the mathematics domain, which is different from the reasoning and SWE domains this paper has evaluated EvoMAS on. Adding this evaluation makes EvoMAS more convincing.

---

> ### Author Rebuttal · Authors · 2026-03-31
>
> We thank the reviewer for recognizing the structured text abstraction as a novel and more robust alternative to direct code generation, and for noting that EvoMAS opens new directions beyond prior agent role generation approaches.
>
> ### 1. Mathematics Domain Evaluation
>
> In Weakness:
> > It would also be interesting to see how would EvoMAS perform on the mathematics domain... Adding this evaluation makes EvoMAS more convincing.
>
> **Response:**
>
> We present preliminary results on AIME (2024 and 2025) using Claude Sonnet 4 as the backbone:
>
> | Method | Accuracy |
> | --- | --- |
> | Single Agent | 43.3% (26/60) |
> | Majority Vote | 48.3% (29/60) |
> | Peer Review | 51.7% (31/60) |
> | Debate | 41.7% (25/60) |
> | **EvoMAS** | **56.7%** (34/60) |
>
> EvoMAS outperforms the best human-designed baseline (Peer Review, 51.7%) by +5.0pp. Analysis of meta-model behaviors shows EvoMAS consistently evolves toward a **Decompose-Solve-Verify topology**, where solution generation and verification are separated and coordinated, building upon patterns from Majority Vote and Peer Review while adapting agent roles to support diverse solution strategies. We will include full results and analysis on a math benchmark in the revision.
>
> ### 2. Query Ordering Robustness
>
> In Question:
> > ...does the order in which the benchmark query is processed first have an impact on the evolution process? How does the paper mitigate it?
>
> **Response:**
>
> As reported in Figure 6(c) in the submission, we evaluate EvoMAS on BBEH-Mini across **11 random query orderings**. Performance concentrates tightly around the default-order baseline of 49.1%, with deviations ranging from -0.9 to +1.9 and a **standard deviation of only 1.1%**. The evolutionary process is robust to query ordering and the reported gains are not artifacts of a particular presentation order.
>
> ### 3. Transfer Analysis
>
> In Question:
> > In Table 6, could the authors discuss more about the reasons EvoMAS can be transferred to other domains after training?
>
> **Response:**
>
> Transferability arises because EvoMAS optimizes system-level structures from cross-query experience, not task-specific solutions:
>
> - With **cross-query experience**, newly generated MAS config may capture more abstract commonality across a family of tasks, which makes it more generalizable.
> - The **initial MAS configuration pool** focuses on general collaborative structures. The derived configurations are also likely to represent generally reusable MAS patterns rather than task-specific ones, potentially making them more generalizable.
> - **Admittedly, tasks with shared structural characteristics transfer best.** As reported in Table 6 in the submission, source tasks involving structured reasoning (Boolean Expressions: 43.7%, Geometric Shapes: 42.6%) transfer more effectively than domain-specific tasks (Linguini: 37.4%), confirming that structural reasoning patterns are the primary vehicle for transfer.
>
> We will expand this analysis in the revision.
>
> ### 4. Tool Evolution
>
> In Weakness:
> > ...agent tooling is an indispensable aspect of multi-agent systems. This paper only evaluated... tool use capability, but did not evaluate how EvoMAS can also improve the tools themselves...
>
> In Question:
> > If the authors were to also consider how tools can be evolved with the MAS, how would EvoMAS incorporate evolving tools in its pipeline?
>
> **Response:**
>
> Tool evolution is an important future direction. Unlike MAS configurations, which are determined before each MAS execution in EvoMAS, tools are dynamically invoked during each agent session. To evolve tools, an LLM/agent-based reviewer can analyze all tool invocations alongside agent trajectories and propose improvements to tool descriptions or implementations, analogous to mutation; a coding agent then implements the changes with a verification loop for reliability. When components across multiple tools are better unified, merging them is analogous to crossover. We see this as a promising future extension of EvoMAS.

---

> > ### Author Rebuttal · Reviewer_MHZo · 2026-03-31
> >
> > Thank you authors for your rebuttals. I am keeping my positive assessment of the work.

---

> > > ### Author Response · Authors · 2026-04-07
> > >
> > > Thank you for your continued positive assessment and for the constructive questions that motivated new experiments. We are glad the AIME results, query ordering robustness analysis, and transfer ability discussion were helpful in further strengthening the paper. All additional experiments and discussions from this rebuttal will be thoroughly incorporated into the revision.

---

### Official Review · Reviewer_Rs8Z · 2026-03-12

**Soundness:** 3
**Presentation:** 3
**Significance:** 3
**Originality:** 2
**Overall Recommendation:** 4
**Confidence:** 3

**Summary:**

This paper proposes EvoMAS, an evolutionary framework for automatically generating multi-agent systems in a structured configuration space rather than through direct code generation.  Concretely, the method represents an MAS as a configurable DAG over agents, where each agent is defined by its backbone model, prompt, and tool set, and then performs pool-based initialization, LLM-guided mutation/crossover, and memory-augmented reuse across queries.  Experiments on BBEH, SWE-Bench, and WorkBench show consistent gains over single-agent methods, fixed human-designed MASs, and prior automatic MAS generation baselines, while also improving executability and runtime robustness.

**Compliance With Llm Reviewing Policy:**

Affirmed.

**Final Justification:**

I think the author's two rounds of rebuttal addressed my main concerns regarding the initial pool and cost. Therefore, I changed my score to 4 to reflect the same. I suggest that the author incorporate all the results into the final version of the paper to enhance the soundness of the article.

**Key Questions For Authors:**

1. Can the authors provide a controlled comparison where EvoMAS is run without cross-query pool/memory accumulation, or alternatively compare against baselines that are also allowed to accumulate cross-query experience?
2. The method is seeded from a pool of human-designed systems, which seems practically reasonable, but it raises the question of whether EvoMAS is primarily discovering new effective MAS structures or mainly refining strong existing templates. Could the authors quantify performance under weaker or more neutral initialization, for example random seeds, much simpler handcrafted seeds, or seeds drawn from only one template family?

**Limitations:**

yes

**Strengths And Weaknesses:**

**Strengths**
- The proposed framework is reasonably well structured, with a clear evolutionary pipeline including initialization, mutation/crossover, selection, and memory-based reuse across queries.
- The empirical evaluation is fairly comprehensive.
- The configuration-based representation appears to improve executability and runtime stability.

**Weaknesses**
- The paper ignores some important and relevant automated workflow-search methods. AFlow already studies automatic optimization over structured agentic workflows, while EvoFlow further introduces evolutionary search over a population of heterogeneous workflows via retrieval, crossover, mutation, and selection.
- EvoMAS is evaluated in a sequential online setting: after each query, its best configuration is added back into the shared pool, and its memory is reused for future queries. But non-evolutionary baselines are evaluated independently without cross-query information sharing.  This makes the comparison somewhat unfair, when the initial pool is explicitly seeded from prior human-designed cooperative MAS configurations.
- The experimental analysis does not clearly break down how much overhead comes from evolution/search itself versus final execution.

---

> ### Author Rebuttal · Authors · 2026-03-31
>
> We thank the reviewer for acknowledging the well-structured evolutionary pipeline and recognizing that the configuration-based representation improves executability and runtime stability over prior approaches.
>
> ### 1. Workflow Search Baselines
>
> In Weakness:
> > The paper ignores some important and relevant automated workflow-search methods. AFlow already studies automatic optimization over structured agentic workflows, while EvoFlow further introduces evolutionary search...
>
> **Response:**
>
> - AFlow and EvoFlow are not agentic: they optimize static LLM workflow graphs (pure LLM calls only), making them inapplicable to tasks like SWE-Bench. EvoMAS additionally jointly optimizes agent roles, backbone models, prompts, and topology.
> - EvoFlow's repository link is *no longer valid*.
> - We compare AFlow against EvoMAS on BBEH-Mini, where its non-agentic constraint is not a blocker. AFlow (3× ensemble + SC) generates three independent solutions per task and selects by self-consistency. EvoMAS outperforms AFlow:
>
> | Method                   | Claude-3.5-Sonnet | Qwen3-235B | Claude-Sonnet-4.5 |
> | ------------------------ | ----------------- | ---------- | ----------------- |
> | Single Agent             | 33.2%             | 37.8%      | 44.8%             |
> | AFlow (3× ensemble + SC) | 36.2%             | 39.3%      | 46.1%             |
> | **EvoMAS**               | **41.5%**         | **45.4%**  | **52.0%**         |
> ### 2. Cross-query Fairness
>
> In Weakness:
> > EvoMAS is evaluated in a sequential online setting... This makes the comparison somewhat unfair...
>
> In Question:
> > Can the authors provide a controlled comparison where EvoMAS is run without cross-query pool/memory accumulation...?
>
> **Response:**
>
> Cross-query accumulation is **a feature, not an unfair advantage**: it is precisely the capability prior MAS methods lack. To isolate its contribution, we disable pool and/or memory updates:
>
> | Setting                                | BBEH-Mini Acc. |
> | -------------------------------------- | -------------- |
> | Single Agent (Qwen3-235B)              | 36.6%          |
> | Peer Review (Qwen3-235B)               | 39.8%          |
> | **EvoMAS (no pool update, no memory)** | 40.4%          |
> | EvoMAS (no pool update, with memory)   | 43.7%          |
> | EvoMAS (with pool update, no memory)   | 42.6%          |
> | **EvoMAS (full)**                      | **49.1%**      |
>
> Without any accumulation, EvoMAS still outperforms both baselines, so **per-query search alone helps**. Enabling memory or pool updates yields further gains, and the pool generalizes beyond the training distribution (Table 6 in the submission).
>
> ### 3. Cost Breakdown
>
> In Weakness:
> > The experimental analysis does not clearly break down how much overhead comes from evolution/search itself versus final execution.
>
> **Response:**
>
> Per ask, we provide BBEH-Mini breakdown:
>
> | Phase               | Avg. Time      | Avg. Tokens  |
> | ------------------- | -------------- | ------------ |
> | Meta-model          | 115.2s (10.6%) | 73K (7.7%)   |
> |   - Meta: Selection | 17.0s (1.6%)   | ~3K (0.3%)   |
> | MAS evaluation      | 212.5 (19.6%)  | 352k (37.0%) |
> | MAS execution       | 756.6s (69.8%) | 527k (55.4%) |
> | *Total per query*   | ~18 min        | 952K         |
>
> For recurring tasks, the cost amortizes: evolve once on a subset and reuse the pool (Transfer Ability and Table 6 in the submission; SWE-Bench-Lite → Verified: ~4% drop). The pool stabilizes after ~300-400 queries, reducing inference to selection plus a single MAS run.
>
> ### 4. Pool Initialization Sensitivity
>
> In Question:
> > ...whether EvoMAS is primarily discovering new effective MAS structures or mainly refining strong existing templates. Could the authors quantify performance under weaker or more neutral initialization...?
>
> **Response:**
>
> A weaker pool hurts evolution performance, but EvoMAS still outperforms the single-agent baseline consistently. Fixed MAS (e.g., Debate) without evolution is inconsistent across domains, whereas EvoMAS with a base pool adapts to each. Two further findings:
>
> 1. From Table 7 in the submission, adding some low-performing configs to the pool (e.g., CRoTo) can degrade performance (44.1% vs. 49.1% at T=3 on BBEH-Mini), indicating pool quality must be balanced against diversity.
> 2. We additionally test the **empty pool** (extremely weak pool) with no pre-designed MAS structures.
>
> | Initial Pool | Evolution Steps | BBEH-Mini Acc. | WorkBench Acc. |
> | ------------ | --------------- | -------------- | -------------- |
> | Single Agent | -               | 29.1%          | 40.1%          |
> | Empty        | 1               | 30.2%          | 42.3%          |
> | Empty        | 3               | 31.5%          | 43.1%          |
> | MAS - Debate | -               | 35.9%          | 33.2%          |
> | Base Pool    | 1               | 45.2%          | 46.2%          |
> | Base Pool    | 3               | **49.1%**      | **48.9%**      |

---

> > ### Author Rebuttal · Reviewer_Rs8Z · 2026-04-04
> >
> > Thank you for the detailed rebuttal and the additional ablations. The new results are helpful, but I still have two important concerns.
> >
> > 1. I remain concerned about the method’s sensitivity to the initial pool.  The paper already states that EvoMAS is seeded with human-designed cooperative MAS configurations, and that these seeds play an important role in the search process.  In the rebuttal, this concern seems only partially alleviated.  In particular, the authors note that adding a low-performing configuration such as CRoTo can noticeably hurt performance, and the empty-pool results also show a clear drop relative to the base pool.  These results suggest that EvoMAS may depend on the quality and composition of the initial pool.  So it may need to carefully choose different seed templates for different task domains.
> >
> > 2. My concerns about cost are still not fully resolved. The paper explicitly defines the reward as a cost-penalized objective, where performance is balanced against execution cost through the coefficient β. The rebuttal provides a useful breakdown of EvoMAS’s time and token usage, but it remains unclear how this cost compares against simpler strong baselines under the same budget. For example, I would like to better understand whether the authors compared against stronger single-agent variants like mini-swe-agent, and whether EvoMAS still provides a clear advantage under a matched token budget. And the paper would benefit from a clearer discussion of how β is chosen and how sensitive the results are to this parameter.

---

> > > ### Author Response · Authors · 2026-04-07
> > >
> > > Thank you for acknowledging our rebuttal and the new ablation results, and for the opportunity to address your two remaining concerns. We deeply appreciate your continued engagement and thoughtful questions.
> > >
> > > ### 1. Initial Pool
> > >
> > > The reviewer's question about pool sensitivity is insightful, raising a key question: what are the requirements on initialization for an iterative method to be practically applicable without extensive manual tuning of the initial pool?
> > >
> > > We address this with a principled argument. A useful iterative method should:
> > > 1. reliably improve upon its initialization;
> > > 2. generalize well across domains from a broadly applicable initialization;
> > > 3. provide a simple methodology to adapt initialization to new domains of specific characteristics.
> > >
> > > These properties hold for successful iterative methods across machine learning (neural network training, clustering, RL), none of which achieve uniformly high performance under arbitrary initializations. EvoMAS satisfies all three, and likewise requires appropriate initialization to fully realize its potential.
> > >
> > > **Evidence for EvoMAS:**
> > >
> > > 1. **Consistent improvement regardless of pool.** Tables 1 and 6 in the submission show EvoMAS consistently improves over all pooled methods. We also test (1) a modified base pool with perturbed agent descriptions and role combinations, (2) a near-degenerate single-agent seed, and (3) three weaker agent-generated pools. Results on BBEH-Mini below confirm EvoMAS improves in all cases; stronger initialization yields higher performance, as expected. `Best in Pool` is the best MAS configuration with the best backbone model.
> > >
> > > | Initial Pool | Best in Pool | EvoMAS | Gain |
> > > | --- | --- | --- | --- |
> > > | Good (base pool, 3 configs) | 42.9% (debate) | 49.1% | +6.2pp |
> > > | Good variations (base pool with perturbed variations) | 42.2% (debate variant) | 49.8% | +7.6pp |
> > > | Overly complex (4 configs, incl. CRoTo) | 42.9% (debate) | 44.1% | +1.2pp |
> > > | Weaker (3 agent-generated configs, *new*) | 41.4% | 45.9% | +4.5pp |
> > > | Weaker (5 agent-generated configs, *new*) | 42.8% | 47.4% | +4.6pp |
> > > | Semi-degenerate (single-agent seed, *new*) | 36.6% (single) | 40.7% | +4.1pp |
> > > | Degenerate (empty pool) | 0% (none) | 33.5% | +33.5pp |
> > >
> > > 2. **Initial pool generalization across domains.** The same base pool (Peer Review, Debate, Majority Vote) performs well across three domains without modification: general reasoning (BBEH), agentic task completion (WorkBench), and mathematics (AIME; see §1 of our response to Reviewer MHZo). Defining a broadly applicable initial pool is itself a contribution.
> > > 3. **Methodology to adapt initialization.** The pool collects representative MAS frameworks from prior literature: a principled methodology, not hand-crafting. For coding, adding MetaGPT and ChatDev from prior work suffices, and EvoMAS achieves strong performance on SWE-Bench-Verified.
> > >
> > > *(New)* **Agent-generated initialization.** The pool can also be generated by an agent: as shown in the table, with 5 agent-generated configs, EvoMAS achieves 47.4%, only 1.7pp below the base pool, confirming both the feasibility of automated initialization and EvoMAS's robustness to pool choice.
> > >
> > > ### 2. Computational Cost
> > >
> > > We show EvoMAS's ability to leverage more computation for accuracy unachievable by baselines via a budget-matched comparison on SWE-Bench-Verified, extending Best-of-6 (Reviewer oxg6 §1).
> > >
> > > Since single agents and simple MAS stop early, leaving the budget unused (Single SWE-Agent: 2.3M; Majority Vote: 11.7M), we force full utilization via: (1) **loop**: re-invoke the agent until the budget is exhausted, with pass@N evaluation; (2) **Best-of-N**: run N independent instances and select the best via the same LLM-as-judge meta-model as EvoMAS.
> > >
> > > | Method | Tokens | Solved Rate |
> > > | --- | --- | --- |
> > > | *Main setting* | | |
> > > | Single SWE-Agent | 2.3M | 33.6% |
> > > | Majority Vote | 11.7M | 30.3% |
> > > | mini-SWE-agent (loop) | 29M | 30.4% |
> > > | SWE-agent (loop) | 29M | 33.6% |
> > > | SWE-agent (Best-of-6) | 24M | 37.0% |
> > > | SWE-agent (Best-of-10) | 40M | 37.8% |
> > > | **EvoMAS** | 29M | **42.7%** |
> > > | *Claude 4.5 Sonnet* | | |
> > > | Single SWE-Agent | 2.7M | 70.4% |
> > > | mini-SWE-agent (loop) | 31M | 70.8% |
> > > | SWE-agent (loop) | 31M | 71.4% |
> > > | **EvoMAS** | 31M | **79.1%** |
> > >
> > > Scaling a single agent (looping, voting, or best-of-N) yields diminishing returns even at budgets exceeding EvoMAS's. EvoMAS outperforms Best-of-10 SWE-Agent (40M tokens) with only 29M, and maintains a consistent +7~8pp advantage under Claude 4.5, confirming the gain is structural and model-agnostic.
> > >
> > > **Impact of β**
> > >
> > > β modulates token usage vs. accuracy: larger β penalizes cost more heavily. Performance is stable (1.1pp band from 10⁻⁶ to 10⁻⁸), with 10⁻⁶ and 10⁻⁷ giving the best accuracy-cost balance, confirming EvoMAS is insensitive to β.
> > >
> > > | β | BBEH-Mini Acc. | Tokens |
> > > | --- | --- | --- |
> > > | 10⁻⁴ | 45.4% | 709K |
> > > | 10⁻⁵ | 46.1% | 723K |
> > > | 10⁻⁶ | 49.1% | 952K |
> > > | 10⁻⁷ | 49.8% | 974K |
> > > | 10⁻⁸ | 48.7% | 996K |

---

### Official Review · Reviewer_Tud1 · 2026-03-20

**Soundness:** 3
**Presentation:** 3
**Significance:** 4
**Originality:** 2
**Overall Recommendation:** 4
**Confidence:** 3

**Summary:**

This paper formulates multi-agent system (MAS) generation as a structured configuration generation problem. It applies evolutionary mechanisms including mutation and crossover to explore new configurations while leveraging memory-based reuse for exploitation. Experiments across benchmarks in diverse benchmarks and domains demonstrate improved task resolved rate and accuracy.

**Compliance With Llm Reviewing Policy:**

Affirmed.

**Final Justification:**

The authors have addressed my main concern regarding differentiation from existing methods. I updated my score to 4.

**Key Questions For Authors:**

- It would be helpful if the authors clarify why comparisons with the recent MAS design methods mentioned in Weakness were not included.
- Although the proposed method shares experimental settings with prior work, would it also work in a zero-shot setting, where only a single task from a new domain is given?

**Limitations:**

yes

**Strengths And Weaknesses:**

Strengths
- The paper formulates MAS configuration as structured text generation to enable flexible and task-adaptive system design.
- It achieves strong performance on a variety of benchmarks/domains such as coding, reasoning, and tool-use.
- It provides extensive analysis to justify key design choices and robustness/sensitivity of components.

Weakness
- Novelty and differentiation: Several works have investigated the automatic design of MAS, and the evolutionary mechanisms have also been explored in EvoAgent [1]. Therefore, while the proposed method presents a well-integrated framework in overall, the novelty of its main components seems limited. Additionally, the paper would benefit from a clearer clarification of how it differs from recent work on automatic design of MAS, including MASS [2] (already cited in the paper but not explained/differentiated enough) and ARG-Designer [3].

References

[1] Evoagent: Towards automatic multi-agent generation via evolutionary algorithms, NAACL'25.

[2] Multi-agent design: Optimizing agents with better prompts and topologies, ICLR'26.

[3] Assemble Your Crew: Automatic Multi-agent Communication Topology Design via Autoregressive Graph Generation, AAAI'26.

---

> ### Author Rebuttal · Authors · 2026-03-31
>
> We thank the reviewer for recognizing the strong cross-domain performance across coding, reasoning, and tool-use, and the extensive analysis justifying key design choices and component robustness.
>
> ### 1. Differentiation from Related Work
>
> In Weakness:
> > ...the evolutionary mechanisms have also been explored in **EvoAgent**. Therefore, while... the novelty of its main components seems limited. ...the paper would benefit from a clearer clarification of how it differs from... **MASS** (already cited in the paper but not explained/differentiated enough) and **ARG-Designer**.
>
> In Question:
> > It would be helpful if the authors clarify why comparisons with the recent MAS design methods mentioned in Weakness were not included.
>
> **Response:**
>
> - EvoMAS's core novelty is **jointly evolving** the full MAS structure (topology, model assignment, tool configuration, and agent components as a unified system). Unlike **EvoAgent**, which evolves single agents (prompt/role/skill) and applies an ensemble-like strategy, EvoMAS optimizes inter-agent coordination structure, making EvoMAS MAS-specific rather than a population of independent agents. EvoMAS also introduces cross-query MAS-pool updates and experience memory. **EvoAgent** is included as a baseline in our main results (Tables 1-4 in the submission): EvoMAS outperforms EvoAgent by +10.5pp on BBEH-Mini and +7.1pp on WorkBench.
> - **MASS** optimizes MAS prompts and topology in separate stages, covering neither model assignment nor tool configuration. It also lacks cross-query learning, which EvoMAS does for the config pool and execution memory. As an ICLR 2026 paper, MASS had not been accepted by peer review before the ICML 2026 submission deadline; we were therefore not required to include it as a baseline. Furthermore, no open-source implementation is available, making replication infeasible within the rebuttal period. We will add a qualitative comparison in the revision.
>
> | Method            | Optimization Target | Optimization Space                               | Joint Evolution | Cross-Query Learning  |
> | ----------------- | ------------------- | ------------------------------------------------ | --------------- | --------------------- |
> | MASS              | MAS (staged)        | Prompt + topology (restricted)                   | ✗ (stage-wise)  | ✗                     |
> | ARG-Designer      | Topology only       | Communication graph for fixed roles              | ✗               | ✗                     |
> | **EvoMAS (Ours)** | **Full MAS system** | **Agents + topology + models + tools + prompts** | **✓ (joint)**   | **✓ (pool + memory)** |
>
> - **ARG-Designer** formulates topology construction as a one-shot autoregressive graph generation problem, mapping queries to communication structures with agents from a fixed role pool. It relies solely on offline training, with no adaptation at inference time. We compare on BBEH-Mini, adapting their code to use Claude Sonnet 4 as the generator backbone and smolagents as the framework (their original GNN-based generator requires trained weights **not released** by the authors, and BBEH falls outside its supported domains):
>
> | Method | Claude-3.5-Sonnet | Qwen3-235B | Claude-Sonnet-4.5 |
> | --- | --- | --- | --- |
> | Single Agent | 33.2% | 37.8% | 44.8% |
> | ARG-Designer | 31.1% | 41.1% | 47.2% |
> | **EvoMAS** | **41.5%** | **45.4%** | **52.0%** |
>
> ARG-Designer underperforms the single agent on Claude-3.5-Sonnet (31.1% vs. 33.2%), confirming that topology-only optimization without joint role and prompt adaptation is unreliable. EvoMAS consistently outperforms ARG-Designer across all backbones. We will include ARG-Designer in the full comparison and expand the related work in the revision.
>
> ### 2. Zero-shot Setting
>
> In Question:
> > ...would it also work in a zero-shot setting, where only a single task from a new domain is given?
>
> **Response:**
>
> Yes. We run EvoMAS with **pool and memory updates disabled** (so each task is independent): evolution still occurs per-query (selection → mutation → evaluation), but discovered configurations are not saved back and no experience is accumulated. This isolates the per-query evolutionary benefit:
>
> | Setting | Pool Updates | Memory Updates | BBEH-Mini Acc. |
> | --- | --- | --- | --- |
> | Single Agent (Qwen3-235B) | ✗ | ✗ | 36.6% |
> | Peer Review (Qwen3-235B) | ✗ | ✗ | 39.8% |
> | EvoMAS (no pool update, no memory) | ✗ | ✗ | 40.4% |
> | EvoMAS (no pool update, with memory) | ✗ | ✓ | 43.7% |
> | EvoMAS (with pool update, no memory) | ✓ | ✗ | 42.6% |
> | **EvoMAS (full)** | **✓** | **✓** | **49.1%** |
>
> Even without pool or memory updates, EvoMAS achieves 40.4%, outperforming both baselines.
>
> For new domains, the "Transfer Ability" section in the submission shows that pools evolved on one task domain retain up to 90% of directly-optimized accuracy **applied zero-shot** to new tasks (e.g., Movie Recommendation → BBEH-Mini: 44.1% vs. 49.1% full evolution).

---

> > ### Author Rebuttal · Reviewer_Tud1 · 2026-04-03
> >
> > Thanks for the rebuttal. The main concerns were sufficiently addressed, especially regarding differentiation from related work. The additional results on the zero-shot setting also look promising. Therefore, I update my score to Weak accept.

---

> > > ### Author Response · Authors · 2026-04-07
> > >
> > > Thank you for confirming that the differentiation from EvoAgent, MASS, and ARG-Designer has been sufficiently clarified, and for finding the zero-shot ablation results promising. We appreciate your recognition that addressing these comparisons strengthened the paper's contribution. All new results and discussions from this rebuttal will be reflected in the revised paper.

---

### Official Review · Reviewer_BY6Q · 2026-03-23

**Soundness:** 3
**Presentation:** 3
**Significance:** 3
**Originality:** 3
**Overall Recommendation:** 4
**Confidence:** 4

**Summary:**

This paper introduces EvoMAS (Evolutionary Generation of Multi-Agent Systems), a framework that automates the design of collaborative multi-agent systems by evolving structured configurations rather than generating executable code. Unlike prior methods that rely on rigid templates or brittle code synthesis, EvoMAS represents agents as declarative YAML specifications (defining roles, prompts, models, tools, and topology) executed by a lightweight interpreter. The system employs an LLM-driven evolutionary process with feedback-conditioned mutation and crossover operators to iteratively refine a pool of candidate configurations seeded from human-designed systems. Key innovations include heterogeneous model assignment (dynamically assigning different backbone models to specific agent roles) and a memory-based reuse mechanism that transfers successful patterns across tasks. Evaluated on diverse benchmarks (BBEH, SWE-Bench, WorkBench), EvoMAS consistently outperforms both human-designed MAS (e.g., MetaGPT, Peer Review) and automatic generation baselines (e.g., AutoAgents, EvoAgent), achieving state-of-the-art results (e.g., 79.1% on SWE-Bench-Verified with Claude-4.5) while maintaining significantly higher execution reliability (~98% success rate vs. <70% for code-generation methods).

**Compliance With Llm Reviewing Policy:**

Affirmed.

**Final Justification:**

Having carefully considered the authors' rebuttal and engaged in thorough discussion, I now fully appreciate the paper's novelty and contributions. The responses have effectively clarified the key technical distinctions and addressed my initial concerns. I am convinced that the work presents a significant advancement in the field and meets the high standards of ICML. The methodology is sound, and the empirical results are compelling. Given the clear articulation of the innovative aspects during this final phase, I believe the manuscript is well-positioned for publication. Therefore, I recommend acceptance and support the decision to include this work in the conference program.

**Key Questions For Authors:**

1.How does the computational cost of the evolutionary search phase (multiple generations per task) compare to the inference cost of the final optimized configuration? Is the trade-off justified for single-shot tasks, or is this approach best suited for recurring task types?
2.The system relies on an LLM-as-a-judge for reward signals. How sensitive is the evolution process to judge biases or hallucinations? Could adversarial examples or tricky tasks fool the judge into selecting suboptimal configurations?
3.Can EvoMAS effectively evolve configurations for open-ended or creative tasks where a clear scalar reward (accuracy/resolved) is unavailable? How would the framework adapt to subjective evaluation metrics?

**Limitations:**

The authors acknowledge limitations regarding the reliance on proprietary models, the scope of cooperative tasks (excluding adversarial settings), and the potential computational overhead of the search process. They also note that the current implementation focuses on specific agent frameworks (SmolAgent, SWE-Agent), though the backend abstraction allows for extensibility. The discussion on safety implications of evolving autonomous agents (e.g., emerging deceptive strategies) is brief and could be expanded. The environmental impact of running extensive evolutionary searches is not deeply analyzed.

**Strengths And Weaknesses:**

Strengths: The configuration-based paradigm is a major conceptual advance, effectively decoupling system structure from execution logic to eliminate syntax errors and runtime failures common in code-generation approaches. The empirical results are robust and impressive, demonstrating clear superiority across reasoning, coding, and tool-use domains, particularly with the automatic LLM selection feature which leverages model heterogeneity. The analysis of emergent behaviors (e.g., role specialization, topology compression, sparse debate graphs) provides deep insights into how evolution discovers efficient architectures that differ qualitatively from human designs. The scaling analysis shows that EvoMAS effectively utilizes increased test-time compute, unlike static baselines that plateau.
Weaknesses: Rigor: The reliance on proprietary models (Claude-4.5, Qwen3-Coder) for the meta-controller and agent backbones limits reproducibility for researchers without API access. The LLM-as-a-judge reward signal, while shown to be effective, introduces potential bias and noise compared to ground-truth metrics, though ablation studies suggest this impact is manageable. Originality: While evolutionary algorithms are well-established, their application to structured configuration space for MAS is novel; however, the core operators (mutation, crossover) remain conceptually similar to standard genetic algorithms, with the "intelligence" largely offloaded to the LLM meta-model. Significance: The computational cost of running evolutionary search for every task (or batch of tasks) might be prohibitive for real-time applications, despite the efficiency of the resulting configurations. The paper focuses on cooperative settings and does not address adversarial or mixed-motive agent interactions.

---

> ### Author Rebuttal · Authors · 2026-03-31
>
> We thank the reviewer for recognizing the "major conceptual advance", the useful behavior analysis (role specialization, topology compression), and EvoMAS's strong test-time compute scaling unlike static baselines.
>
> ### 1. Reproducibility
>
> In Weakness:
> > The reliance on proprietary models... limits reproducibility...
>
> **Response:**
>
> Our framework is agent-backend agnostic (Section 4.1 in the submission), supporting open-weight models for both backbone and meta-model. With Qwen3-235B as meta-model, EvoMAS achieves 58.2% on SWE-Bench-Verified vs. 63.8% with Claude Sonnet 4 (Reviewer oxg6 §3). We will also open-source EvoMAS.
>
> ### 2. Novelty of Evolutionary Operators
>
> In Weakness:
> > ...the core operators (mutation, crossover) remain conceptually similar to standard genetic algorithms, with the "intelligence" largely offloaded to the LLM meta-model.
>
> **Response:**
>
> We agree mutation and crossover are analogies from genetic algorithms and are shown to work well empirically. The novelty lies in formalizing MAS as structured configurations (agents, prompts, models, tools, and topology), jointly evolving this unified structure via LLM-driven search, and cross-query pool updates and experience memory, absent from prior MAS methods.
>
> ### 3. Computational Cost and Scope
>
> In Weakness:
> > The computational cost... might be prohibitive for real-time applications...
>
> In Question:
> > How does the computational cost of the evolutionary search phase compare to the inference cost of the final optimized configuration? Is the trade-off justified for single-shot tasks, or is this approach best suited for recurring task types?
>
> **Response:**
>
> BBEH-Mini breakdown:
>
> | Phase             | Avg. Time      | Avg. Tokens  |
> | ----------------- | -------------- | ------------ |
> | Meta-model        | 115.2s (10.6%) | 73K (7.7%)   |
> | - Meta: Selection | 17.0s (1.6%)   | ~3K (0.3%)   |
> | MAS evaluation    | 212.5 (19.6%)  | 352k (37.0%) |
> | MAS execution     | 756.6s (69.8%) | 527k (55.4%) |
> | *Total per query* | ~18 min        | 952K         |
>
> MAS execution dominates (~70% time, ~55% tokens); overhead comes from evaluation and meta-model.
>
> For single-shot tasks, EvoMAS leverages a higher compute budget for accuracy otherwise unachievable (Reviewer oxg6 §1; Reviewer Tud1 §2).
>
> For recurring tasks, the cost amortizes: evolve once on a subset and reuse the pool (Transfer Ability and Table 6 in the submission; SWE-Bench-Lite → Verified: ~4% drop). The pool stabilizes after ~300-400 queries, reducing inference to selection plus a single MAS run.
>
> ### 4. Reward Signal Reliability
>
> In Weakness:
> > The LLM-as-a-judge reward signal... introduces potential bias and noise..., though ablation studies suggest this impact is manageable.
>
> In Question:
> > How sensitive is the evolution process to judge biases or hallucinations? Could adversarial examples or tricky tasks fool the judge into selecting suboptimal configurations?
>
> **Response:**
>
> Table 8 in the submission shows the LLM-judge vs. ground-truth gap is only +3.4% on BBEH-Mini and +3.1% on SWE-Bench-Verified. We also provide a judge model ablation (agreement = % of verdicts matching oracle):
>
> | Benchmark | Judge Model | Agreement | Performance |
> | --- | --- | --- | --- |
> | BBEH-Mini | Claude Sonnet 4.5 | 95.4% | 50.4% |
> | " | Claude Sonnet 4 (default) | 93.0% | 49.1% |
> | " | Qwen3-235B (open-weight) | 90.4% | 47.3% |
> | SWE-Bench-Verified | Claude Sonnet 4.5 | 97.2% | 64.4% |
> | " | Claude Sonnet 4 (default) | 96.6% | 63.8% |
> | " | Qwen3-235B (open-weight) | 92.8% | 62.4% |
>
> All judges achieve >90% agreement; Qwen3-235B drops accuracy by only 1.8pp / 1.4pp on BBEH-Mini / SWE-Bench-Verified. Robustness comes from trace-level evaluation: structural failures are detectable without ground-truth labels, and we observed no noticeable biases or hallucinations. We agree adversarial attacks on MAS evolution are an important topic as agent learning shifts to LLM-based evolution.
>
> ### 5. Scope: Open-ended Tasks and Adversarial Settings
>
> In Question:
> > Can EvoMAS effectively evolve configurations for open-ended or creative tasks where a clear scalar reward is unavailable? How would the framework adapt to subjective evaluation metrics?
>
> In Weakness:
> > The paper focuses on cooperative settings and does not address adversarial or mixed-motive agent interactions.
>
> **Response:**
>
> *On open-ended tasks:* EvoMAS uses a scalar reward from an LLM-as-judge reviewing results and trajectories across multiple aspects (correctness, completeness, reasoning quality). For open-ended tasks, the judge adapts to task-appropriate dimensions, with the LLM determining relevant aspects and aggregating scores.
>
> *On adversarial settings:* We focus on collaborative settings as they are most widely used in general-purpose agents. EvoMAS has no structural constraint here: mixed-motive agents can be represented as structured MAS configurations, and we see this as an important future direction.
>
> ---
>
> We'll expand limitation & impact.

---

> > ### Author Rebuttal · Reviewer_BY6Q · 2026-04-06
> >
> > I have carefully reviewed the authors' responses and found them to be comprehensive and convincing. I acknowledge the clarification regarding the backend-agnostic nature of the framework and the commitment to open-source release, which sufficiently addresses the reproducibility concerns. The detailed cost breakdown provides valuable context for understanding the computational trade-offs, and the evidence supporting the reliability of the LLM-as-a-judge framework is robust. I agree with the authors' reasoning on the novelty of formalizing MAS configurations and their evolutionary approach. The explanations regarding the scope of open-ended tasks and future directions are logical. Overall, the authors have successfully addressed my concerns and strengthened their arguments. I believe the paper makes a significant contribution to the field, and therefore, I am in favor of accepting this submission.

---

> > > ### Author Response · Authors · 2026-04-07
> > >
> > > Thank you for carefully reviewing our responses and for confirming that all concerns have been addressed. We are glad the backend-agnostic clarification and open-source commitment addressed the reproducibility concern, and that the cost breakdown and judge agreement ablation across three models were convincing. Everything introduced during this rebuttal will be fully incorporated into the final version.
> > >
> > > 🔹 Should the rebuttal have further strengthened your view of the work, we would be honored if you felt a score adjustment was warranted, and we sincerely appreciate your time and thoughtful engagement with our work.

---

### Decision · Program_Chairs · 2026-04-30

**Decision:**

Accept (regular)

**Comment:**

The reviewers found the paper to make a solid contribution overall. A central strength is the formulation of multi-agent system design as structured configuration evolution rather than direct code generation, which improves executability, robustness, and search flexibility. Reviewers also found the empirical evaluation strong across diverse domains, and the rebuttal substantially strengthened the paper by clarifying differentiation from related MAS design methods, providing cost and equal-budget comparisons, and showing that the gains are not simply due to higher compute or a particular meta-model choice.

There are still limitations, including nontrivial search cost and some dependence on initialization quality, and the core evolutionary ingredients are not individually radical. That said, the paper presents a clear, effective, and well-supported framework for automatic MAS design, with strong results and useful analysis. On balance, I recommend acceptance.